# Controls on the hydraulic geometry of alluvial channels: bank stability to gravitational failure, the critical-flow hypothesis, and conservation of mass and energy

Jon D. Pelletier

Department of Geosciences, The University of Arizona, 1040 East Fourth Street, Tucson, Arizona 85721–0077, U.S.A.

*Correspondence to*: Jon D. Pelletier (jdpellet@email.arizona.edu)

**Abstract.** The bankfull depths, widths, depth-averaged water velocities, and along-channel slopes of alluvial channels are approximately power-law functions of bankfull discharge across many orders of magnitude. What mechanisms give rise to these patterns is one of the central questions of fluvial geomorphology. Here it is proposed that the bankfull depths of alluvial channels are partially controlled by the maximum heights of gravitationally stable channel banks, which depend on bank material cohesion and hence on clay content. The bankfull depths predicted by a bank-stability model correlate with observed bankfull depths estimated from the bends in the stage-discharge rating curves of 387 U.S. Geological Survey gaging stations in the Mississippi River Basin. It is further proposed that depth-averaged water velocities scale with bankfull depths as a result of a self-regulatory feedback among water flow, relative roughness, and channel-bed morphology that limits depth-averaged water velocities to be within a relatively narrow range associated with Froude numbers that have a weak inverse relationship to bankfull discharge. Given these constraints on channel depths and water velocities, bankfull widths and along-channel slopes consistent with observations follow by conservation of mass and energy of water flow.

## 1 Introduction

The bankfull depths, $h$, widths, $w$, depth-averaged water velocities, $v$, and along-channel slopes, $S$, of alluvial channels exhibit power-law relationships with bankfull discharge, $Q$:

$$h \propto Q^k, w \propto Q^b, v \propto Q^m, S \propto Q^z \qquad (1)$$

where $k \approx 0.4$, $b \approx 0.5$, $m \approx 0.1$, and $z \approx -0.4$ (Leopold and Maddock, 1953). Many studies have proposed a process-based understanding of these patterns (see reviews by Ferguson (1986) and Singh (2003)), but none has achieved widespread acceptance.

Schumm (1960) documented that alluvial channels with sand-rich bed and bank material tend to be wide and shallow, while alluvial channels with silt-and-clay-rich bed and bank material tend to be narrow and deep. Schumm's findings have led nearly all subsequent researchers to consider the resistance of bank material to fluvial erosion to be the key factor controlling alluvial channel width (e.g., Parker, 1979; Eaton and Millar, 2004; Dunne and Jerolmack, 2020). Bank retreat, however, is also driven by gravitational failure (ASCE, 1999), a process that limits bank heights to values that depend on bank material cohesion and

hence on clay content. The gravitational failure of channel banks may partially control bankfull depths via a self-regulatory mechanism in which channel incision and/or floodplain deposition tend to increase bank height, triggering bank failure when a critical bank height, dependent on bank material cohesion, is exceeded (Andrews, 1982), thus introducing new sediment into the channel bed that, as it is redistributed by fluvial processes, tends to reduce the channel depth back towards a critical value. This paper demonstrates that bankfull depths predicted by a bank-stability model correlate with observed bankfull depths

estimated from the bends in the stage-discharge rating curves of 387 U.S. Geological Survey (U.S.G.S.) gaging stations in the Mississippi River Basin (MRB). This analysis supports the hypothesis that the gravitational failure of channel banks partially controls bankfull depths and complements the recent work of Dong et al. (2019) that documented a relationship between bank-material texture and the hydraulic geometry of alluvial channels in the Selenga River Delta.

Grant (1997) proposed a critical-flow hypothesis in which depth-averaged water velocities are self-regulated via interactions

between the water flow and the channel-bed morphology. Grant (1997) argued that in steep ($\gtrsim 0.01$ m m$^{-1}$) channels the Froude number rarely exceeds one for extended periods of time due to interactions between the free surface and the bed that result in an approximate balance between forces that accelerate the flow and forces that extract energy from the flow. Such a balance may also extend to coarse-bedded channels, in which the water flow is prone to wave drag associated with flow around bed sediment grains that protrude above the water surface (Wohl, 2013). Wave drag can be expected to be more common in coarse-

bedded channels relative to channels with finer bed sediments both because they have relatively large bed roughness elements that more readily protrude above the water surface and a tendency towards shallower flows as a result of their characteristically large width-to-depth ratios (Schumm, 1960).

Central to the critical-flow hypothesis is the existence of a self-regulatory feedback in which an increase in velocity is met with an increase in drag that tends to reduce the velocity and hence the Froude number. Similarly, a decrease in velocity tends

to be met with a decrease in drag that tends to increase the velocity. Here it is hypothesized that such a self-regulatory feedback is not limited to steep channels. In less-steep, sand-bedded channels, faster flow tends to facilitate the development of larger and more well-developed bedforms (which tend to form at Froude numbers ~0.1-1 (e.g., Simons and Richardson, 1966)) that increase relative roughness and hence drag. The mechanisms of self-regulation and the Froude numbers at which steep and less-steep channels may achieve this self-regulation thus differ, but both are likely to have self-regulatory interactions between

the flow and the bed that limit the Froude numbers of bankfull discharges. Consistent with this hypothesis, here it is documented that bankfull Froude numbers, and hence the ratio of depth-averaged water velocities to the square root of bankfull depths, tend to be within a relatively narrow range that has a weak inverse relationship to bankfull discharge.

The bankfull widths of alluvial channels are set by the requirement that channels convey geomorphically effective water discharges. Conservation of mass, together with the clay-content control of bankfull depths and the Froude-number control of

water velocities, thus constrains bankfull widths.

Conservation of energy constrains along-channel slopes. The conversion of gravitational potential energy into the kinetic energy of water leads to a relationship among along-channel slopes, bankfull Froude numbers, bankfull depths, and the size of

the largest bed roughness elements, which in gravel-bedded channels tend to be bed sediment grains and in sand-bedded channels tend to be ripples and/or dunes.

## 2 Methods

### 2.1 Controls on bankfull depths

The maximum stable height, $h_c$, of an alluvial channel bank subject to gravitational shear failure is proportional to bank-material cohesion, $C$ (Taylor, 1937; Terzaghi and Peck, 1967; Hunter and Schuster, 1968; Chen et al., 1969; ASCE, 1999):

$$h_c = \frac{N_s}{\rho g} C, \tag{2}$$

where $\rho$ is the bulk density of the bank material, $g$ is the acceleration due to gravity, and $N_s$ is a stability parameter dependent on the geometry of the potential failure surface (e.g., planar, log-spiral, or circular), the pore pressure of the bank material (which is governed by the water table position if the pore pressure is assumed to be hydrostatic), and the angles of the bank and of internal friction (see Table 1 for a list of variables).

In order to estimate a reference $N_s$ value appropriate for understanding how gravitational stability may influence the scaling
of alluvial channel bankfull depths to discharge, a steep bank (i.e., near-vertical at the top of the bank but decreasing to approximately 45˚ near the toe) with an internal friction angle of 35˚ (typical for a loamy or clayey sand), near-saturated conditions, and a log-spiral potential failure surface were assumed. Near-saturated conditions are consistent with the fact that gravitational shear failure has been documented to occur most frequently during the falling limbs of flood discharges when pore pressures tend to be associated with near-saturated conditions (e.g., Casagli et al., 1999; Simon et al., 2000). Chen et al.
(1969) derived $N_s$ values for prescribed angles of the bank and of internal friction for unsaturated conditions. For a friction angle of 35˚, $N_s$ values in Table 1 of Chen et al. (1969) decrease with increasing bank angle from $N_s$ = 22 for a 60˚ bank to $N_s$ = 12 for a 75˚ bank and $N_s$ = 7.5 for a vertical bank. Hunter and Schuster (1968) limited their analysis to cases with no internal friction (hence their absolute $N_s$ values are not applicable here) but documented an approximately 3-fold decrease in $N_s$ values from unsaturated conditions (i.e., $M = h_w \gamma_w / h_c \gamma' \approx 1$, where is $h_w$ is the depth to the water table below the top of the bank, $\gamma_w$
is the unit weight of water, and $\gamma'$ is the submerged unit weight of the bank material) to near-saturated conditions (i.e., $M = 0$). Combining the results of Chen et al. (1969) and Hunter and Schuster (1968) suggests that $N_s$ values for a saturated bank with an internal angle of friction of 35˚ vary from approximately 7.3 for a 60˚ bank to $N_s \approx 4$ for a 75˚ bank and 2.5 for a vertical bank.

Bank material cohesion varies linearly from 0 (for cohesionless sand) to approximately 90 kPa (for pure clay) for moisture
contents in the range of 7 to 40% according to a least-squares linear regression of the data from Dafalla (2013) (Fig. S1):

$$C \approx (900 \pm 70) p_c. \tag{3}$$

where the units of $C$ is Pa and $p_c$ is percent. The uncertainty in Eq. (3) is the standard error (i.e., one standard deviation) resulting from the regression.

Combining Eqns. (2) and (3) and assuming a bulk density of 1700 kg m$^{-3}$ and a value of $N_s \approx 6$ (corresponding to a near-
saturated bank with an angle of approximately 65°, i.e., an average angle for a bank that is near-vertical at the top and decreases
to an angle of approximately 45° near the toe) yields

$$h_c \approx 0.35 p_c. \tag{4}$$

Absent site-specific data for bank angles, the largest source of uncertainty in the proportionality coefficient in Eq. (4) as applied
to specific locations is likely the bank angle, since relatively modest variations in bank angle (e.g., from 90° to 75°) are
associated differences in $N_s$ of approximately a factor of 3, i.e., much larger than other sources of uncertainty such as that
between cohesion and clay content as quantified by Eq. (3). Section 4 provides discussion on how uncertainty in bank angle
and other factors such as bank vegetation limit the precision of Eq. (4) to specific locations. The primary objective of this
paper, however, is to document an increase, on average, in bankfull channel depth with increasing bank-material clay content:

$$h \approx 0.35 \, p_c. \tag{5}$$

assuming that bankfull depth is approximately equal to the maximum gravitationally stable bank height.

To test the tendency for channel depth to increase, on average, with increasing bank-material clay content as predicted by
Eq. (5), the bankfull depths for 387 U.S.G.S. gaging stations in the MRB were estimated using the bends in the stage-discharge
rating curves (Fig. S2) for each station. Predictions of bankfull depth using Eq. (5) were then compared to the observed bankfull
depths derived from the rating curves. The gNATSGO soil database (Soil Survey Staff, 2019) was used to estimate the percent
clay content of the floodplain deposits adjacent to each station. This analysis focuses on the MRB because there is no readily
available soils database for any other continental-scale river basin that resolves floodplains in comparable detail and is based
on a similar richness of field-based soil texture measurements.

The bankfull depth for each U.S.G.S. gaging station was estimated using the intersection of the linear regressions of peak
annual gage height (used as a proxy for stage) to peak annual discharge obtained using the five smallest and five largest
discharges in each record (shown as gray circles in the example of Fig. S2). This intersection, or bend, in the stage-discharge
rating curve can identify the stage and discharge above which overbank flow occurs (Copeland et al., 2000), provided that the
slope of the high-flow linear regression is much smaller than the slope of the low-flow linear regression, which is a signature
of the abrupt widening of flows as they transition from in-channel to overbank.

The analysis presented here began by including data from all U.S.G.S. gaging stations in the MRB with available peak
discharge data (U.S. Geological Survey, 2020). Only those stations for which the slope of the high-flow linear regression is at
least five times smaller than the slope of the low-flow linear regression were retained. In addition, only those stations that had
at least 20 years of data, have a contributing area larger than 100 km$^2$, are not located close to major infrastructure (based on
an inspection of each station location in Google Earth imagery), and have a resulting bankfull depth of greater than 2 m were
retained. Channels with bankfull depths less 2 m were removed because such channels tend to be associated with low clay
contents that are inherently difficult to estimate in the field (see Sect. 3.1 for more detail on the potential bias associated with
estimating low clay contents). In order to further filter out stations where the low-flow linear regression is potentially

unrepresentative of the hydraulic behavior of in-channel discharges, only those stations for which the extrapolation of the low-flow linear regression is close to zero flow depth at zero discharge were retained. Stations for which the low-flow regression does not extrapolate to a flow stage close to zero at zero discharge may have gage height data that are not an accurate proxy for flow stage and/or have other data quality issues that preclude an accurate estimate of bankfull depth using the stage-discharge rating curve. What represents "close" should not be based on an absolute error, e.g., 0.5 m, because such a criterion would require that the low-flow regressions for deep channels be relatively more accurate than those for shallow channels. Here "close" was defined as being within 50% of the bankfull stage from zero. That is, if the bankfull stage is 5 m, then the extrapolation of the low-flow regression to zero discharge must yield a stage within 2.5 m of zero in order for that station to be retained in the analysis. Similarly, if the bankfull stage is 2 m, then the extrapolation of the low-flow fit to zero discharge must be within 1 m of zero. A total of 387 stations met these criteria.

To estimate the floodplain clay content for each of the 387 U.S.G.S. gaging stations, the depth-averaged percent clay content from soil depths of 5 to 150 cm was computed for every pixel within the MRB using gNATSGO. These depths were chosen to avoid the soil horizon close the surface (typically the O and/or A horizon, which may have clay contents reflective of surficial biological processes that are not representative of the rest of the profile) and because soil properties at depths greater than 150 cm can be inconsistently encoded in U.S. soil databases (Miller and White, 1998). A moving geometric mean (averaging distance of 10 km) of percent clay content was computed within floodplains mapped by Nardi et al. (2019). Because some U.S.G.S. gaging stations are located in channels with narrow floodplains that are not resolved in Nardi et al. (2019), the Nardi et al. (2019) floodplain map was augmented with single-pixel-width valleys defined by pixels with contributing areas larger than 100 km$^2$ following a steepest-descent routing of contributing area within the National Map Digital Elevation Model (DEM) (Archuleta et al., 2017). Bankfull depths predicted by Eq. (5) were then compared to observed bankfull depths using a Pearson correlation coefficient, a root-mean-squared error (RMSE), the percentage of values correctly predicted to within a factor of 2, and a $p$-value that quantifies the likelihood of the null hypothesis that the predicted and observed bankfull depths may be correlated by chance.

To assess the potential impact of errors in measurements of percent clay content on predictions of the maximum stable bank height and therefore of bankfull depth using Eq. (5), synthetic predictions for bankfull depths, $h_{pred,syn}$, were generated equal to 0.35 times samples of synthetic percent clay content, $p_{c,syn}$, drawn from a lognormal distribution designed to mimic the distribution of bankfull depths of U.S.G.S. gaging stations in the MRB, plus a normally distributed random error with a mean of zero and standard deviation of σ:

$$h_{pred,syn} = 0.35\big(p_{c,syn} + \sigma\eta\big) \tag{6}$$

where η is a normally distributed random variable with a mean of zero and a standard deviation of 1. For σ = 0, Eq. (6) produces synthetic data precisely equal to Eq. (5). With finite values of σ, Eq. (6) produces synthetic data with scatter that can be used to assess how errors in percent clay content may affect the relationships between observed and predicted bankfull depths.

**2.2 Controls on depth-averaged water velocities and bankfull widths**

The critical-flow hypothesis implies that bankfull Froude numbers, $F$, are limited to a relatively narrow range of values with an upper limit close to 1 for gravel-bedded channels and a similarly narrow but somewhat lower range of values for sand-bedded channels. In Sect. 3.2 it is demonstrated that this variation in $F$ can be quantified using a power-law relationship between $F$ and $Q$:

$$F \propto Q^n. \tag{7}$$

The power-law exponents reported in this paper were determined via a least-squares linear regression of the logarithms of the data. The definition of Froude number provides a linkage among depth-averaged bankfull water velocities, bankfull Froude numbers, and bankfull depths:

$$v = F\sqrt{gh}. \tag{8}$$

Equations (7) and (8) thus constrain the value of $m$ to be

$$m = \frac{k}{2} + n. \tag{9}$$

The exponent $b$ in the relationship between bankfull width and discharge can be constrained by conservation of mass of water assuming steady, uniform flow, consistent with many previous models for the downstream hydraulic geometry of alluvial channels (e.g., Lindley, 1919; Smith, 1974; Ferguson, 1986; Huang et al., 2004; Julien, 2014):

$$b = 1 - k - m. \tag{10}$$

**2.3 Controls on along-channel slopes**

The Darcy-Weisbach equation

$$v = \sqrt{\frac{8ghS}{f}}, \tag{11}$$

is based on conservation of energy for steady, uniform flow, i.e., that the gravitational potential energy per unit mass, $ghS$, produces a depth-averaged water velocity associated with a drag force per unit mass exerted by the channel bed on the water equal to $(f/8)v^2$, where $f$ is the friction factor (e.g., Ferguson, 1986). Equation (11) can be rewritten as

$$S = \frac{F^2}{8}f. \tag{12}$$

Here the Variable Power Equation (VPE) of Ferguson (2007) is used to quantify $f$:

$$(8/f)^{1/2} = a_1 a_2 \beta^{-1} / \left(a_1^2 + a_2^2 \beta^{-5/3}\right)^{1/2} \tag{13}$$

where $a_1$ and $a_2$ are coefficients (equal to 6.1 and 2.4 based on the least-squares minimum error for velocity in the calibration performed by Ferguson (2007)), and $\beta$ is the relative roughness. The VPE equation was chosen because it provides a transition between the Manning-Strickler $1/3^{rd}$ power scaling of friction factor to relative roughness for $\beta \gg 1$ (nearly all sand-bedded channels and some gravel-bedded channels) to a quadratic scaling when $\beta$ is $\lesssim 1$ (channels with very coarse beds) that accords well with available data (Ferguson, 2007).

Relative roughness depends on whether or not bedforms are present. Table S2 of Ohata et al. (2017) identifies the range of $F$ and $d_{50}$ values conducive to ripple and/or dune development in alluvial channels. Of the 3790 data points in the Ohata et al.

(2017) dataset, 1574 (42%) have ripples or dunes, of which 19% are in the field and the remaining 81% are in the laboratory. By cross-referencing those results with the Dunne and Jerolmack (D&J) (2018) global dataset (i.e. by drawing a curve in $F$ vs. $d_{50}$ space that separates channels that have ripples and/or dunes from those that do not, and assuming the existence of ripples and dunes in channels of the D&J dataset that have $F$ and $d_{50}$ values that sit above and to the left (Fig. 3B) of the envelope curve separating channels with and without ripples and dunes in the Ohata et al. (2017) dataset), Sect. 3.3 demonstrates that 96% of sand-bedded channels in the D&J global dataset have $F$ and $d_{50}$ values conducive to ripple and/or dune development and therefore have a roughness that is likely dominated by bedforms rather than by bed sediment grains. The D&J global dataset includes 789 observations of $d_{50}$, $S$, and, $h$, and 711 observations of $d_{50}$, $S$, $w$, $h$, and $Q$ drawn from the literature.

For gravel-(and-coarser)-bedded channels, relative roughness is defined in the calibration of Ferguson (2007) as the ratio of 84$^{th}$ percentile of bed grain diameter to the hydraulic radius. Since the analysis of this paper relates Eq. (13) to data from the D&J global dataset that uses $d_{50}$ instead than $d_{84}$, it is assumed that $d_{50} \approx d_{84}/2$ and, because $w > 10h$ for all points in the D&J global dataset, that the bankfull hydraulic radius is approximately equal to the bankfull depth. The relative roughness for gravel-bedded channels, $\beta_g$, in the D&J global dataset can, therefore, be approximated as

$$\beta_g \approx \frac{2d_{50}}{h}. \tag{14}$$

For sand-bedded channels, relative roughness can by estimated as (Bathurst, 1993):

$$\beta_s \approx \frac{1.1H(1-e^{-25\alpha})}{3h}. \tag{15}$$

where $H$ is the bedform height and $\alpha$ is the ratio of bedform height to length, $L$.

Combining Eqs. (12)-(15) gives an equation for the along-channel slopes of gravel-bedded channels consistent with conservation of energy:

$$S_g = \frac{F^2}{a_1^2 a_2^2}\left(\frac{2d_{50}}{h}\right)^2\left(a_1^2 + a_2^2\left(\frac{h}{2d_{50}}\right)^{5/3}\right), \tag{16}$$

and an analogous equation for sand-bedded channels:

$$S_s = \frac{F^2}{a_1^2 a_2^2}\left(\frac{1.1H(1-e^{-25\alpha})}{3h}\right)^2\left(a_1^2 + a_2^2\left(\frac{3h}{1.1H(1-e^{-25\alpha})}\right)^{5/3}\right). \tag{17}$$

The heights and lengths of ripples and dunes can be estimated as (Yalin, 1964):

$$H \approx \frac{h}{6}\left(1 - \frac{\tau_c}{\tau_0}\right), \tag{18}$$

and

$$L \approx 1000d_{50}. \tag{19}$$

Equation (17), therefore, can be rewritten in terms of $d_{50}/h$ for a prescribed value of $\alpha$ as

$$S_s \approx \frac{F^2}{a_1^2 a_2^2}\left(\frac{18000\alpha(1-e^{-25\alpha})}{1.1\left(1-\frac{\tau_c}{\tau_0}\right)^2}\frac{d_{50}}{h}\right)^2\left(a_1^2 + a_2^2\left(\frac{1.1\left(1-\frac{\tau_c}{\tau_0}\right)^2}{18000\alpha(1-e^{-25\alpha})}\frac{h}{d_{50}}\right)^{5/3}\right). \tag{20}$$

# 3 Results

## 3.1 Controls on bankfull depths

Figure 1 illustrates the tendency for the clay contents of floodplain deposits adjacent to many smaller channels in the MRB to be lower than those of larger channels. For example, the North and South Platte Rivers have typical floodplain clay contents
225   <10%, while the Platte River has typical floodplain clay contents of ≈10-20%, and the Missouri and Lower Mississippi Rivers have typical clay contents of ≈20-30%. There are many relatively small channels, however, that have clay contents > 30% due to clay-rich local bedrock. As such, there isn't a precise, one-to-one correlation between clay content and contributing area or discharge, but rather a general tendency for channels conveying larger discharges to have more clay-rich floodplain deposits.

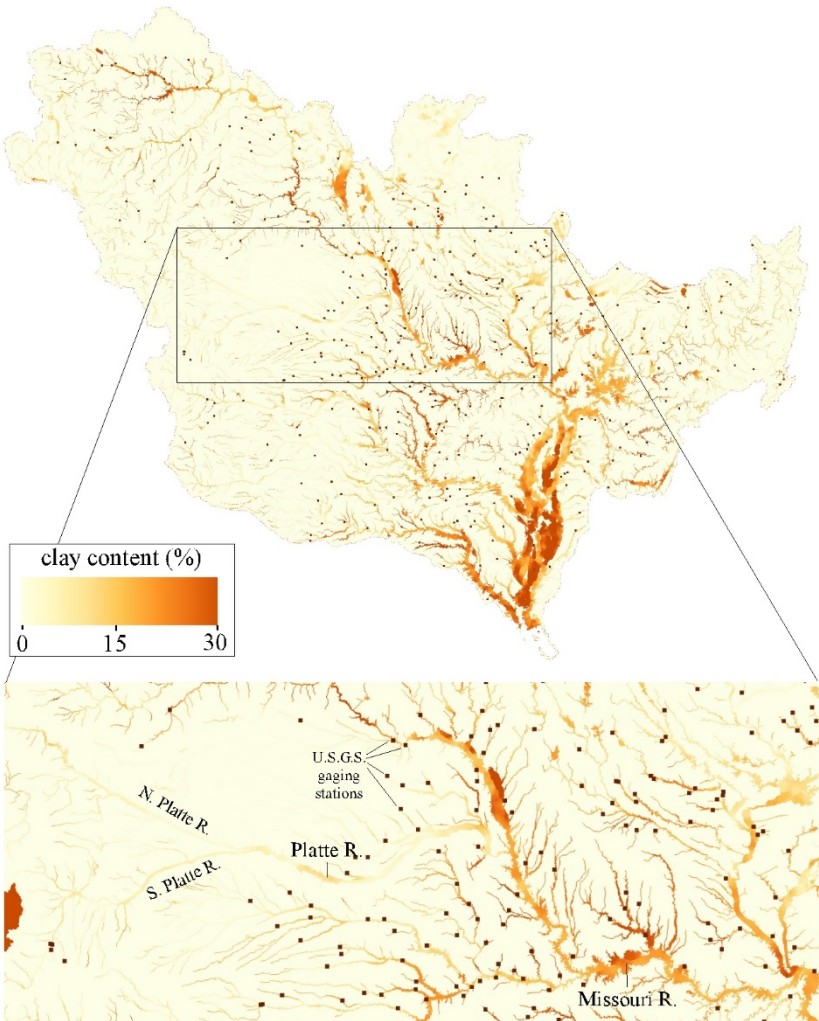

230

**Figure 1: Color map of the average floodplain deposit clay content in the Mississippi River Basin. Also shown are the locations of the 387 U.S.G.S. gaging stations where predicted bankfull depths were computed using Eq. (5) and the observed bankfull depths were estimated based on the bends in the stage-discharge rating curves.**

Figure 2(a) compares the bankfull depths predicted by Eq. (5) to observed bankfull depths estimated from the bends in the stage-discharge rating curves. Predicted bankfull depths have a Pearson correlation coefficient of 0.42 with observed bankfull depths, a RMSE of 1.7 m, and 84% of the data points are within a factor of 2 of the observed bankfull depth. The $p$ value, i.e., the chance that the correlation between $h_{pred}$ and $h_{obs}$ could have occurred by mere chance, is $\sim 10^{-17}$.

Figures 2(b) and 2(c) illustrate the potential impact of errors in percent clay contents on predicted bankfull depths using Eq. (5) with $\sigma = 0.06$ and 0.12 (6 and 12%), respectively. These $\sigma$ values were chosen because, while gNATSGO does not provide an error estimate, a recent soil property dataset created using machine learning algorithms has an estimated RMSE of 12% (Ramcharan et al., 2018) and a value half that size allows for the effect of error size to be assessed. For a relatively small error ($\sigma = 6\%$), there is a spread of values around the 1:1 line, with a larger relative spread for smaller clay contents, i.e., a 6% error results in a 100% relative error for a percent clay content of 6% (i.e., $h_{pred,syn} \approx 2$ m) but a 50% relative error for a percent clay content of 12% (i.e., $h_{pred,syn} \approx 4$ m). As the $\sigma$ value increases to 12%, the spread of values around the 1:1 line increases as expected but $h_{pred,syn}$ values less than approximately 2 meters also appear to be biased upward (i.e., the geometric mean of $h_{pred,syn}$ values deviate from the 1:1 line).

This upward bias may be associated with the difficulty of measuring very low clay contents in the field. Clay contents estimated in the field can only be constrained to be within the range from 0 to 10% clay. If the actual clay content is close to 0 (e.g., < 1%), the clay content estimate is likely to be overestimated by a much larger fraction than would be the case for a larger clay content (e.g., 5% clay is 400% greater than 1% clay but only 50% lower than 10% clay). The result may be an upward bias in clay content for clay contents less than approximately 10%, which, according to Eq. (5), may be associated with channels less than 2-3 m in bankfull depth. Channels with bankfull depths less than 2 m (Sect. 2.1) were excluded from the analysis due to this potential bias.

255

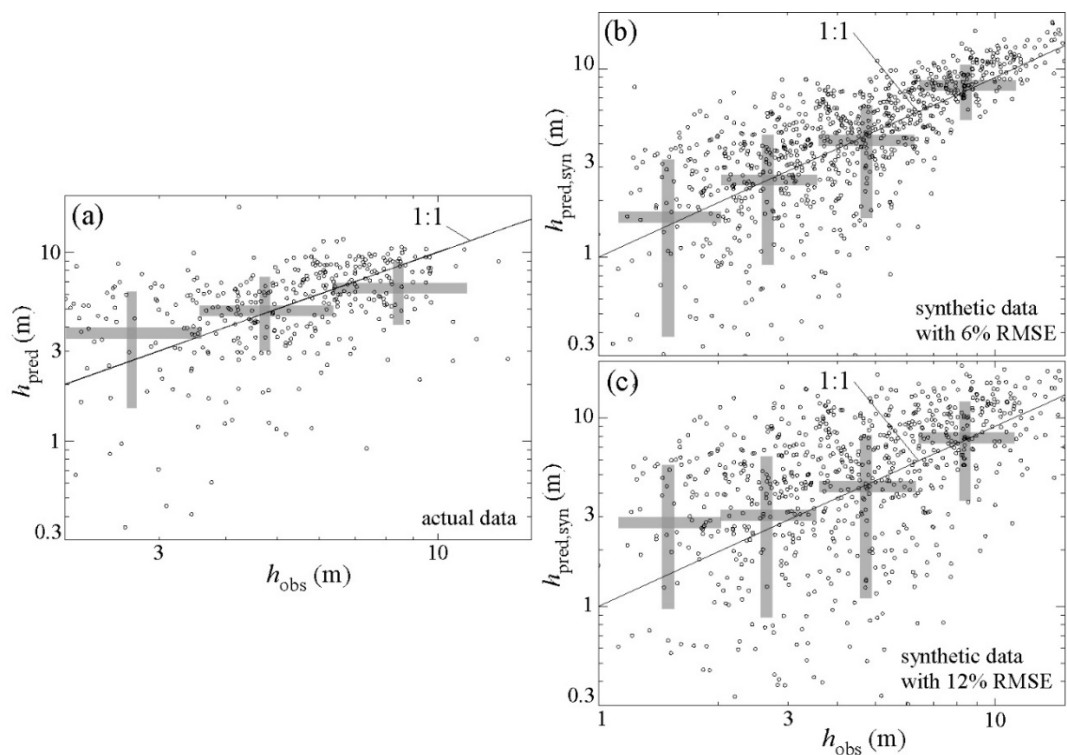

**Figure 2: Plots of predicted bankfull depths, $h_{\text{pred}}$, calculated using Eq. (5) versus observed bankfull depths, $h_{\text{obs}}$, estimated based on the bends in the stage-discharge rating curves. (a) Actual data. (b)&(c) Synthetic data (Eq. (6)) with (b) $\sigma = 6\%$ and (c) $\sigma = 12\%$, respectively. The small open circles are individual data points and the gray rectangles illustrate the geometric means and standard deviations within each logarithmically spaced bin. The gray rectangles are for visualization purposes only – no analyses are performed on these values.**

## 3.2 Controls on depth-averaged water velocities and bankfull widths

Ripples and dunes tend to form at lower Froude numbers in channels with finer bed sediments (Fig. S3). The curve in Fig. S3 used to identify the range of $F$ and $d_{50}$ conditions conducive to ripple and/or dune development is reproduced in Fig. 3, where 96% of sand-bedded channels in the D&J global dataset are below the upper limits of $F$ and $d_{50}$ conducive to ripple and/or dune development identified using the Ohata et al. (2017) dataset. As such, it is assumed for the purposes of this analysis that sand-bedded channels in the D&J global dataset are dominated by bedform roughness while gravel-bedded channels in the D&J global dataset are dominated by grain roughness.

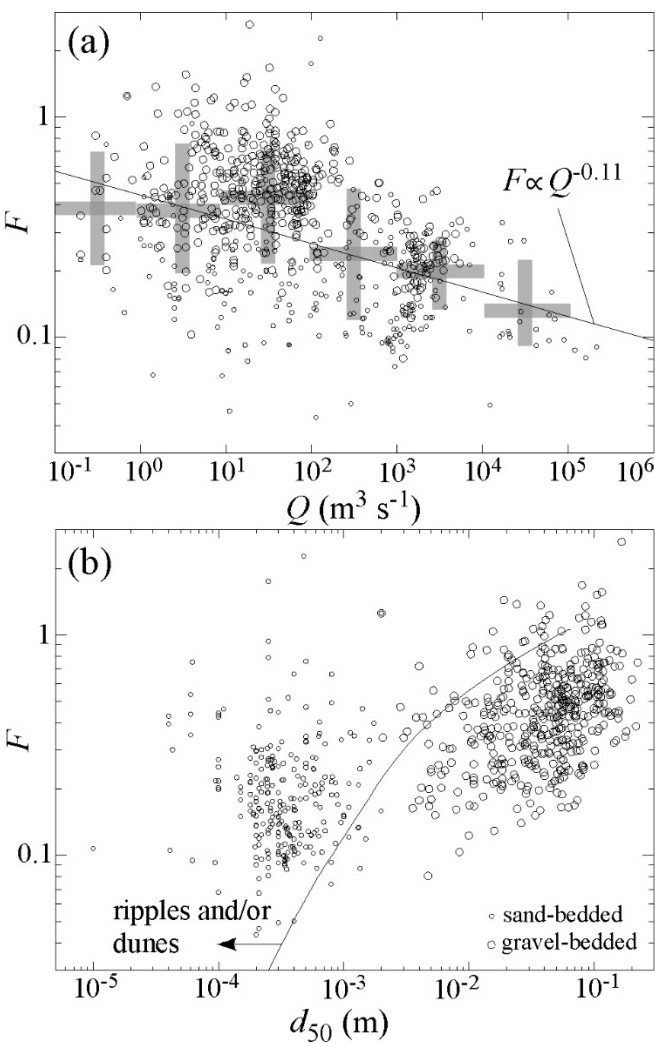

**Figure 3: Bankfull Froude number relationships with bankfull discharge and median bed grain diameter. Plot of bankfull Froude number, *F*, as a function of (a) bankfull discharge, *Q*, and (b) median grain size, $d_{50}$, for the Dunne and Jerolmack (2018) global dataset. Small circles correspond to sand-bedded channels ($d_{50} < 2$ mm), large circles correspond to gravel-bedded channels ($d_{50} > 2$ mm). The curve in (b) defining channels that likely have ripples and/or dunes is based on the subset of 3791 field studies and experiments compiled by Ohata et al. (2017) and graphed in Fig. S3.**

Using the D&J global dataset, alluvial channel depths and widths scale with bankfull discharges to the $0.402 \pm 0.006$ and $0.512 \pm 0.007$ powers, respectively ($R^2 = 0.86$ and $0.89$) (Fig. 4). If we take the $-0.116 \pm 0.009$ Froude-number-discharge scaling exponent obtained from least-squares regression to the logarithms of the data (Fig. 3(a)) and the $0.402 \pm 0.006$ depth-discharge scaling exponent as a starting point, Eqs. (9) and (10) predict a width-discharge exponent of $0.51 \pm 0.01$, i.e., precisely equal to that observed in the D&J global dataset.

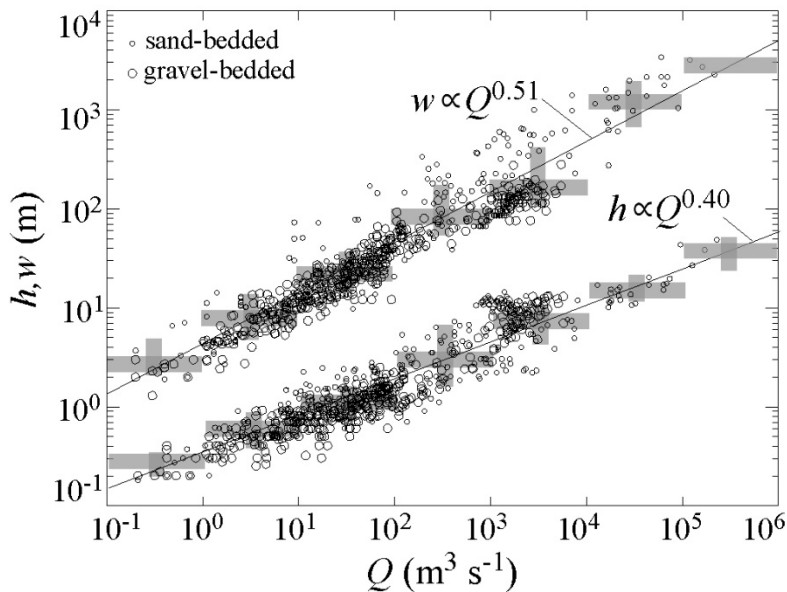

**Figure 4: Plots of bankfull depth, *h*, and width, *w*, as a function of bankfull discharge, *Q*, from the Dunne and Jerolmack (2018) global dataset, along with least-squared linear regressions of the logarithms of the data, indicating *b* = 0.512 ± 0.007 and *k* = 0.402 ± 0.006 for this dataset.**

### 3.3 Controls on along-channel slopes

Along-channel slopes predicted by Eqs. (16) and (20) are consistent with observed values in the D&J global dataset (Pearson correlation coefficient of 0.77) (Fig. 5). For sand-bedded channels, in which ripples and/or dunes are likely to be the dominant roughness elements, the predicted values plotted in Fig. 5 assume $\tau_c/\tau_0 \approx 0$ (consistent with the suspended-load-dominated conditions common in sand-bedded channels (Dade and Friend, 2000)) and a representative $\alpha$ value of 0.05 (based on the range 0.05-0.1 reported by Guy et al. (1966)). Figure S4 illustrates the sensitivity of the predictions of the along-channel slopes of sand-bedded channels to the presence/absence of bedforms and the assumed value of $\alpha$. For alternative scenarios in which a) an unrealistically large value $\alpha$ = 0.25 is assumed, and b) bed grains are assumed to be the dominant roughness elements (i.e., no bedforms are present), Eqs. (16) and (20) predict along-channel slopes that are approximately an order of magnitude above and below observed values, respectively.

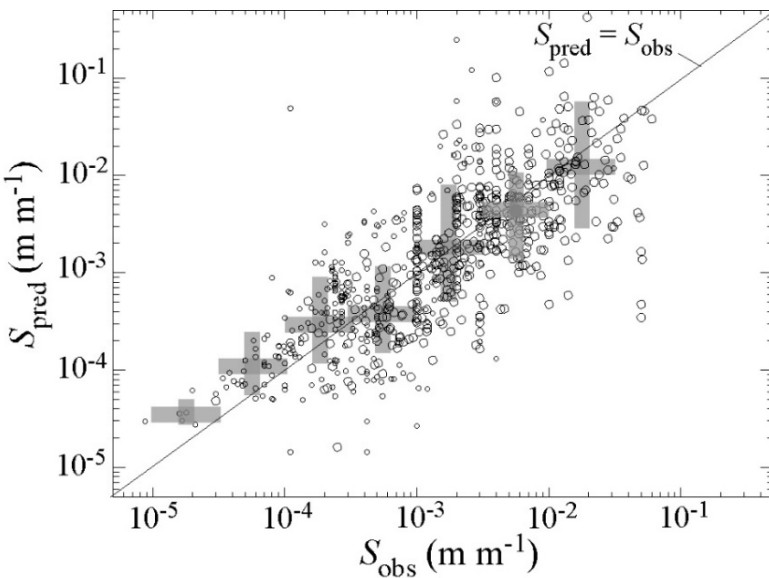

**Figure 5: Plot of predicted along-channel slope, $S_{pred}$, using Eqs. (16) and (20) as a function of observed along-channel slope, $S_{obs}$, in the Dunne and Jerolmack (2018) global dataset. Predicted values for sand-bedded channels assume $\tau_c/\tau_0 \approx 0$ (consistent with the suspended-load-dominated conditions common in sand-bedded channels (Dade and Friend, 2000)) and $\alpha \approx 0.05$ (Guy et al., 1966).**

## 4 Discussion

The model of this paper posits that floodplain deposit clay contents partially control the maximum stable heights of channel banks. This control is unlikely to result in a precise correlation between clay contents and bankfull channel depths for at least two reasons besides data errors. First, the incised depth of a channel that flows through a section of higher clay content to a section of lower clay content may be more strongly controlled by the lower clay content because the downstream reach may act as the local base level of erosion for the upstream reach. Second, alluvial channels can adjust to spatial variations in bank material cohesion by varying the bank angle in addition to the bank height (Knighton, 1974). For a 35° angle of internal friction, for example, a bank angle of 60° has a stability factor, $N_s$, that is approximately three times larger than a vertical bank with the same cohesion (e.g., Fig. 3 of Chen et al. (1969)). As such, an alluvial channel that flows through a section with less cohesive bank material compared to neighboring sections may adopt a less steep bank (thus increasing the stability factor, $N_s$) instead of, or in addition to, becoming wider and shallower in order to minimize variations in channel depth that might otherwise drive large spatial variations in rates of aggradation/incision. Further tests of the model of this paper may require a better understanding of how channel depths and/or bank angles adjust to spatial variations in bank material cohesion through a channel network.

It is also important to consider how potential errors in the data may contribute to the observed scatter in Fig. 2. The estimates of bankfull depths presented here are not exact because gage height (the height of the water above a reference point) is used as a proxy for flow stage. Also, uncertainties in clay content of just 5-10% percent are capable of creating scatter comparable

to that in Fig. 2. Despite such large potential errors and the bias they may introduce into predictions of bankfull depths, the analysis presented here rules out the possibility that floodplain deposit clay contents and bankfull depths are related by chance (i.e., $p \sim 10^{-17}$) and it demonstrates that Eq. (5) predicts bankfull depths to within a factor of 2 of the observed bankfull depths for 84% of the 387 stations included in the analysis.

The model of this paper posits that bankfull depth may be self-regulated via a tendency for an increase in bank height caused by channel incision and/or floodplain deposition to trigger bank failure when a critical bank height, dependent on bank material cohesion, is exceeded. An important role for gravitational failure in controlling alluvial channel geometry is consistent with process-based studies of bank retreat, in which gravitational shear failure has been documented to occur frequently during the falling limbs of flood discharges when pore pressures tend to be highest (e.g., Casagli et al., 1999; Simon et al., 2000). An important role for shear failure in bank retreat is also consistent with Li et al.'s (2015) finding that bankfull depth correlates with fluid viscosity because viscosity controls the permeability of bank material and hence pore pressures and therefore susceptibility to gravitational shear failure. Fluvial erosion of the bank toe is still necessary to remove material slumped from the bank into the channel and likely plays an important role in driving bank retreat in channels with cantilevered banks (Pizzuto, 1994). However, if the bank is more than twice the height of the zone of scour, gravitational shear failure nevertheless will be the process by which the majority of material is removed from the bank (Tao et al., 2019). That said, gravitational failure and fluvial shear stresses act in concert in such a way that identifying which process is dominant may be difficult. Fluvial scour transports slumped material away from the bank, likely keeping bank angles higher than they would be without fluvial scour. Gravitational failure transports material from higher on the bank to the toe (often following fluvial scour, which is maximized at the toe), likely keeping bank angles lower than they would be without gravitational failure. More research is needed, but the relative importance of these two processes may be reflected in the bank angle, i.e., a bank angle persistently much less than vertical may indicate the dominance of gravitational failure while a bank angle persistently at or above 90° (i.e., a cantilever or overhang) may indicate the dominance of fluvial scour.

Bank vegetation plays a significant role in controlling bank stability, but it is unlikely that such control is responsible for the scaling relationships that are the focus of this paper, as such scaling relationships exist across climatic regions with very different vegetation characteristics. In addition, the stability of any bank is primarily a function of its weakest portion or layer, i.e., failure is more likely to occur in a zone of low shear strength compared to a zone of high shear strength, all else being equal. Failure of a weaker zone underlying a stronger zone can create a cantilever, but such cantilevers cannot continue to grow indefinitely, hence rates of long-term retreat in stronger and weaker zones of the same bank will tend to be similar. This, together with the fact that vegetation is likely to strengthen just the uppermost approximately 2 m of channel banks (globally, >99% of roots are found in the uppermost 2 m of the soil (Jackson et al., 1996)), suggests that bank material shear strength is unlikely to be controlled primarily by vegetation in channels deeper than 2 m. A dependence of bank retreat rates on vegetation has been documented in many studies (e.g., Ielpi and Lapôtre, 2020). However, such studies may not fully account for the fact that wetter climates with more vegetation also tend to have more clay-rich soils, leaving open the question of whether it is bank vegetation or material texture that is most responsible for bank resistance to erosion.

The model and analysis of this paper are simplified in at least five specific ways that bear mentioning. First, it does not account for tension cracks that, if present, can lower the maximum stable height below that predicted by Eq. (2) (Darby and Thorne, 1994). Second, it does not account for the role of vegetation in bank stability, which can increase bank heights by at least a factor of two over values predicted using bank material texture alone (Huang and Nanson, 1998). Third, it assumes a maximally incised channel, i.e., a channel that has incised to the point of reaching the threshold of bank stability quantified by Eq. (2). Quaternary climatic changes have driven cycles in which channel aggradation has been followed by a positive feedback of incision and channel narrowing (e.g., Bull, 1991) that have likely made many alluvial channels prone to an incised state. Such considerations, however, do not apply to some types of alluvial channels, e.g., small-scale channels formed in the laboratory and those in which sediment supply is not heavily influenced by climatic changes. Fourth, it involves no explicit constraint on the width-to-depth ratio of alluvial channels despite the fact that such a constraint may play an important role in some cases. An important concept in rill erosion is that unit stream power is maximized for a width-to-depth ratio of $\approx$ 0.5-3, with the specific value dependent on the cross-sectional functional form (e.g., Moore and Burch, 1986). A similar concept may limit incision in channels with small width-to-depth ratios (Huang and Nanson, 2000), reducing the likelihood of channels with $w/h \lesssim 1$ and hence potentially limiting $h$ to values smaller than that set by Eq. (2) for channels with small discharges. Such a control does not seem likely in the channels of the D&J global dataset (since $w/h \gtrsim 10$), but it may play an important role in some types of channels and should be explicitly considered in future research. Fifth, the analysis assumes that the dominant sources of roughness in the channels of the D&J data are ripples/dunes or gravel clasts. Long-wavelength topographic features such as bars and meanders are not likely to be dominant roughness/drag-inducing elements given that the presence/absence of the flow separation that tends to dominate drag depends sensitively on the maximum slope of bedforms and other obstacles to the flow, with slopes in excess of 0.2 m/m generally needed to trigger flow separation (e.g., Lefebvre et al., 2014). Vegetation can certainly be a dominant source of roughness on the beds of ephemeral channels, however, and some of the scatter in the analysis of this paper may be a result of vegetation-induced bed roughness.

More data on the relationship between bank cohesion and bank material texture are needed. This study used clay content as a predictive variable for cohesion in part because a transfer function is available between clay content and cohesion based on the work of Dafalla (2013) used in Figure S1. However, silt content also affects cohesion (Huang et al., 2006) and the findings of Schumm (1960) suggest that both the silt and clay content of bank material are relevant to understanding alluvial channel geometry. The specific surface area of bank material may be a more accurate predictive variable for cohesion than either clay content or silt and clay content (weighted equally), as specific surface area includes both the silt and clay contents but weighs the presence of clays more heavily (Huang et al., 2006).

The analysis presented here used a power-law relationship to quantify the relationship between $F$ and $Q$ (Eq. (7)). However, a scaling break appears to exist in the D&J global dataset (Fig. 3(a)), with $F$ values approximately constant for $Q$ values less than $\sim 10^2$ m$^3$ s$^{-1}$ (i.e., discharges dominated by gravel-bedded channels). This break may be consistent with the critical-flow hypothesis, i.e., steep, predominantly gravel-bedded channels in the D&J global dataset may have $F$ values predominantly in the range of 0.3-1, independent of channel size/discharge, because the general lack of ripples and dunes in gravel-bedded

channels requires that the increase in drag near critical flow conditions be caused primarily by wave drag, while in sand-bedded channels the increase in drag near critical-flow conditions is likely to be caused by ripples and/or dunes that form at a range of Froude numbers weakly dependent on bankfull discharge. It would be useful for future research to investigate the relationships among $F$, $Q$, bed texture, and the presence/absence of ripples and/or dunes to more fully test this hypothesis and its implications for potential breaks in scaling within Eq. (1).

The model of this paper implicitly includes the sediment-supply control on along-channel slope documented by Li et al. (2015), Pfeiffer et al. (2017) and Blom et al (2017). Drainage basins with higher rates of sediment supply erode faster, resulting in coarser sediments being delivered to channels, all else being equal (Attal et al., 2015). Coarser sediments increase along-channel slopes because steeper slopes are necessary to achieve critical or near-critical water velocities in alluvial channels with coarser bed sediments (Eqs. (16)&(20)).

## 5 Conclusions

This paper proposed that the bankfull depths of alluvial channels may be partially controlled by the maximum heights of gravitationally stable channel banks, which depend on bank material cohesion and hence on clay content. Bankfull depths predicted by a bank-stability model correlate with the observed bankfull depths estimated using the bends in the stage-discharge rating curves for 387 U.S.G.S. gaging stations in the Mississippi River Basin. It was proposed, inspired by the critical-flow
hypothesis of Grant (1997), that depth-averaged water velocities scale with bankfull depths as a result of a self-regulatory feedback among water flow, relative roughness, and channel-bed morphology that limits velocities to be within a relatively narrow range associated with Froude numbers that have a weak inverse relationship to bankfull discharge. Given these constraints on channel depths and depth-averaged water velocities, bankfull widths and along-channel slopes consistent with observations follow from conservation of mass and energy of water flow. The model of this paper provides a novel process-
based understanding of the hydraulic geometry of alluvial channel networks. A better understanding of how cohesion relates to bank material clay and/or silt contents, how channel depths and/or bank angles adjust to spatial variations in bank material cohesion through a channel network, and the relationships among bankfull discharges, bankfull Froude numbers, bed texture, and the presence/absence of ripples and/or dunes would enable a more comprehensive test of the model of this paper and a better understanding of the bank-textural controls on the hydraulic geometry of alluvial channels more generally.

*Data Availability* The Supplementary Material contains all of the data used in the paper not published elsewhere.

*Competing interests* The author declares that he has no competing interest.

*Acknowledgements* I wish to thank Gordon Grant, Roberto Fernández, Christopher Hackney, Luke A. McGuire, and Alexander B. Prescott for careful reviews that led to a significantly improved manuscript.

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

Table 1: List of variables

| Symbol | Variable Name/Description | Units | Reference value(s) |
|---|---|---|---|
| $a_1$ | coefficient in VPE model | | 6.1 |
| $a_2$ | coefficient in VPE model | | 2.3 |
| $\alpha$ | ratio of ripple or dune height to length | | 0.05 |
| $b$ | exponent in power-law scaling of bankfull width and discharge | | 0.51 |
| $\beta$ | roughness ratio | | |
| $\beta_g$ | roughness ratio for gravel-bedded channels | | |
| $\beta_s$ | roughness ratio for sand-bedded channels | | |
| $C$ | bank material cohesion | kPa | 0-100 |
| $d_{50}$ | median bed grain diameter | mm | |
| $\eta$ | normally distributed random variable | | |
| $f$ | friction factor | | |
| $F$ | bankfull Froude number | | |
| $g$ | acceleration due to gravity | m s$^{-2}$ | 9.81 |
| $\gamma_w$ | unit weight of water | kN m$^{-3}$ | 9.81 |
| $\gamma'$ | submerged unit weight of soil | kN m$^{-3}$ | 10 |
| $h$ | bankfull channel depth | m | |
| $h_{pred}$ | bankfull channel depth predicted by Eq. (5) | m | |
| $h_{pred,syn}$ | synthetic bankfull channel depth predicted by Eq. (6) | m | |
| $h_{obs}$ | observed bankfull channel depth | m | |
| $h_c$ | maximum stable bank height | m | |
| $h_w$ | depth of water table below top of bank | m | |
| $H$ | bedform height | | |
| $k$ | exponent in the power-law relationship between bankfull depth and discharge | | 0.40 |
| $L$ | bedform spacing | | |
| $m$ | exponent in the power-law relationship between velocity and bankfull discharge | | 0.1 |
| $M$ | saturation parameter in Hunter and Schuster (1968) | | 0-1 |
| $n$ | exponent in the power-law relationship between bankfull Froude number and bankfull discharge | | -0.11 |
| $N_s$ | stability parameter in bank mass failure equation | | 6 |
| $p_c$ | percent clay content in bank material | | |
| $p_{c,syn}$ | synthetic percent clay content in bank material | | |
| $Q$ | bankfull discharge | m$^3$ s$^{-1}$ | |
| $\rho$ | bank material bulk density | kg m$^{-3}$ | 1700 |
| $S$ | along-channel slope | | |
| $\sigma$ | standard deviation of synthetic percent clay content | | 0.06, 0.12 |
| $\tau_c$ | boundary shear stress threshold for entrainment | Pa | |
| $\tau_0$ | boundary shear stress | Pa | |
| $v$ | depth-averaged water velocity | m s$^{-1}$ | |
| $w$ | bankfull channel width | m | |
| $z$ | exponent in the power-law relationship between along-channel slope and bankfull discharge | | -0.4 |

535