# Peer review of "Controls on the hydraulic geometry of alluvial channels: bank stability to gravitational failure, the critical-flow hypothesis, and conservation of mass and energy"

_Earth Surface Dynamics, 2020_

## Referee Comment (RC1) · Gordon Grant (Referee) · 22 Jul 2020

This intriguing paper suggests that the well-known downstream hydraulic geometry relations and exponents can be at least partially explained by postulating that bankfull height (hence bankfull depth) is set by the balance of forces leading to gravitational failure. Key to this mechanism is the amount of bank cohesion, which is interpreted to result from clay content. A further constraint on hydraulic geometry comes from the recognition that Froude numbers in natural channels are typically less than or near-critical, giving rise to a relatively well-behaved relationship between Froude number

and discharge, from which the velocity exponent (and therefore the width exponent by conservation of mass) can be calculated. An extension of this analysis allows the longitudinal slope exponent to be calculated as well. Results using this approach compare quite favorably with large published datasets in both the Mississippi basin and globally.

That said, I remain somewhat skeptical of both the approach and the results. My skepticism is rooted in both some of the underlying theory and assumptions used to derive the mathematical relations that underpin this work, and also the wide variance and assumptions in the data used to validate the methods. I cannot put my finger on a single "smoking gun" amidst all of this, but I was quite surprised that given all the assumptions and uncertainties, the predicted relationships were almost spot-on the widely published values (starting with Leopold and Maddock) for the hydraulic geometry relationships. Herein lies the intrigue, but I'm not convinced that the right answer was obtained for the right reasons. To his credit, the author acknowledges the fragility of some of the assumptions, and calls for additional work to better understand key relationships. Thus, this paper will stimulate discussion and work and deserves to be shared with the broader community, irrespective of any misgivings.

The postulated relation between bankfull height and clay content builds on previous work, but as the author points out, the assumption in this earlier work was that cohesion limited erosion by increasing the required near-bank shear stresses for fluvial entrainment. Here the focus is on gravitationally-driven stress failures, modeled with a very simple 1-D relationship (Eq. 2), that when parametrized, gives rise to an even simpler linear equation between bankfull depth and maximum gravitationally stable bank height. This relation is tested at the scale of the Mississippi River basin by comparing bankfull height derived from USGS gage station data with predicted height calculated from clay content derived for each station using gNATSGO soil survey data and estimated with a 10km moving window. Not surprisingly, the resulting scatter plot is. . .well, scattered with a weak positive trend (Fig. 2A). The author does synthetically consider what the error in this relationship might look like; this error analysis is not propagated

through the rest of the analysis, however. At the end of the day, I remain skeptical that broad-scale soil survey data can be used to parametrize clay content for point data (USGS stations). To be fair, in the discussion the author does consider other sources of potential error beyond data error, including the use of stage data as a proxy for bank height, and other possible channel adjustments to varying clay content (i.e., varying bank angle, widening). But taken together, these uncertainties raise doubts about the validity of the paper's central claim that bankfull height is primarily controlled by gravitational slope failure.

While I'm appreciative of the author's invoking the "critical flow hypothesis" (correct citation is Grant, 1997, not 2000 as it appears in text but not references), I'm somewhat confused by the role it plays in this story. As I understand it, the author argues that this hypothesis suggests that the range of Froude numbers for both sand and gravel-bed channels should be limited to near- or less than 1.0, and that therefore Froude number and discharge should be weakly and inversely related. The logic here is not entirely clear, and the mechanism that restricts Froude numbers is not entirely accurate. In its original form, the hypothesis argues that in steep channels (typically S>0.01), interactions between the free surface (particularly hydraulic jumps) and the bed result in a rough balance between forces that accelerate the flow and forces that extract energy from the flow and thus retard it, thus promoting near-critical flow conditions. This condition applies irrespective of grain size, although the author rightly points out that the actual bed features that set up this interaction are different for sand, gravel, and even boulder bed channels. A recent paper tests this idea in the flume and articulates the mechanism well (Piton and Recking, 2019). Most of the streams in the Dunne and Jerolmack data set have slopes much less than 0.01, and consequently much lower Fr, as the data shown in the paper point out.

More to the point, in my view, the primary control on Fr is channel slope, and I suspect that this is what is behind the weak inverse correlation between Fr and discharge (Fig. 3A). The simple theoretical dependency between Fr and S is shown in Grant (1997;

**ESurfD**
Fig. 4); a more sophisticated treatment is given in Palucis and Lamb (2017; Fig. 2). Since smaller streams tend to be steeper than larger ones, my hunch is that discharge is more of a proxy for slope than a driver as in Fig. 3A. This slope dependency is also probably lurking behind the grain-size/Fr relationship in Fig. 3B as well. None of this fundamentally invalidates the argument being made by the author but I think these relations should be acknowledged, as they have bearing on the physical mechanisms underlying the presentation.

Overall, although the tone of this review might seem negative, I think the paper is well-written and the author raises some interesting points. This work should stimulate a provocative exchange and deserves a wider audience and discussion after these technical issues are addressed.

Gordon Grant

References cited:

Grant, Gordon E. "Critical flow constrains flow hydraulics in mobile‐bed streams: A new hypothesis." Water Resources Research 33.2 (1997): 349-358.

Palucis, M. C., and M. P. Lamb. "What controls channel form in steep mountain streams?." Geophysical Research Letters44.14 (2017): 7245-7255.

Piton, Guillaume, and Alain Recking. "Steep Bedload‐Laden Flows: Near Critical?." Journal of Geophysical Research: Earth Surface 124.8 (2019): 2160-2175.

---

## Referee Comment (RC2) · Roberto Fernandez (Referee) · 16 Aug 2020

**1   Summary**

The manuscript presents a combination of approaches to identify the controls on the hydraulic geometry of alluvial rivers. It includes hydraulic and geotechnical consider-ations which is, in my opinion, a good and relevant approach and likely to improve our understanding of bankfull geometry controls. The manuscript first presents a

geotechnical-based approach to compare the maximum stable height of a riverbank and compares the results with over 300 data points obtained from the Mississippi River Basin. It then analyzes data from the literature to establish correlations between bankfull width and depth as functions of bankfull discharge. Finally, the analysis presents two equations to predict along-channel slopes for sand-bedded and gravel-bedded rivers and compares them with observed values. In general, the manuscript shows good agreement between predicted and observed values. The author closes by recognizing some of the shortcomings of the approach and highlighting research needs to improve our understanding of bankfull geometry controls.

**2  General comments:**

After carefully reading the manuscript, I have the following comments, which I hope the author will find relevant and useful.

1. The use of the data by Dafalla (2013) seems misrepresented in this manuscript (Figure S1). Cohesion values reported by Dafalla (2013) correspond to pure sand, sand and clay mixtures with 5

In addition, Dafalla (2013) shows that for the same clay content (15

I really believe that the approach proposed in this manuscript has a lot of potential and others have included geotechnical considerations in models for stream restoration (e.g. CONCEPTS, see Langendoen et al. 2001; RVR Meander, see Motta et al. 2012). I would encourage the author to dig out some more references regarding cohesion estimates for soils that are more relevant for the Mississippi River Basin. For example, Masada (2009) presents a very extensive report on geotechnical parameters for the state of Ohio and includes different relations between sediment properties (cohesion for example) and soil composition (amount of silt, amount of clay, etc.). Their approach is specific to highway embankments but the results of their tests might be more general

in terms of cohesion values in relation to clay contents.

2. The use of equation (2) might not be appropriate for riverbanks. The use of that equation as presented by Chen (1969) and Terzaghi et al. (1996; p. 271-272) is for soil embankments located above the water table. Several authors have used it in the past as discussed by ASCE (1988) but even there, the authors suggest that critical depth approaches are not accurate when the most common bank failure mechanisms for riverbanks are due to tension cracks that cause toppling or cantilever failures.

Assuming the equation is indeed an appropriate approach for riverbanks, I would encourage the author to explore the sensitivity of its input variables to other values. Chen (1969) shows a wide range of Ns values that depend on the internal friction angle of the material (which is sensitive to moisture content) and the actual slope of the bank. The smallest stable bank height would be given by the smallest possible safety parameter Ns so why not explore a range of Ns values. When the channel has low flow, the bank might be quite dry and its maximum stable height would be quite different from that obtained with a saturated bank (e.g. during the falling limb of a hydrograph where the river stage is getting lower but the bank remains saturated). It would be very useful to see these considerations in the analysis. The author discusses the issue briefly but more details regarding bank failure mechanisms and their prevalence might strengthen the manuscript.

3. Sensitivity analysis: Figures 2b and 2c present results for bank heights based on a synthetic dataset. If the author estimated clay contents using averaging windows for a soils dataset, why not extract second order statistics from it and use them directly instead of creating a synthetic dataset?

4. Use of the Mississippi River Basin data: The author clearly states why the MRB data are used. However, not knowing much about the many different locations along the basin, I have a few questions. (1) What percentage of the cross sections analyzed can be considered natural? (2) Did the author discard those locations where the navigable

channels are maintained by the US Army Corps of Engineers? (3) Of the many stations used, how many might be influenced by river control structures (dams, wing dams, chevrons, etc.) or road infrastructure (e.g. culverts, bridges)?

5. Figures 3 and 4: It is not at all clear why the author includes regression plots of the Dunne and Jerolmack (2018) dataset. Based on the abstract and introduction, it was unexpected that a different dataset appears in the manuscript and becomes the focus of the second half. I understand the use of the dataset for Figure 5, which is new but the content of Figures 3 and 4, is not. I would encourage the author to make it clear to the reader earlier that the DJ dataset is a substantial part of the analysis and to state explicitly the novelty of including figures 3 and 4.

6. Figure 5: I have a few specific questions about the analysis leading to Fig. 5. (1) What is the number (and percentage) of cases that report ripples/dunes over the entire Ohata (2017) dataset? (2) For those reporting ripples/dunes, what is the number and percentage of measurements obtained in the laboratory and in the field? (3) For those in the field, how many are for large rivers? Cisneros et al. (2020) show that traditional dune scaling equations overestimate the size of dunes in large rivers and propose the following relation between dune height (H) and water depth (h) – H   0.056h - 0.12h. (4) Are the only sources of roughness in the DJ data the ripples/dunes or gravel size? What about bars, meandering, vegetation?

7. As a final general comment, I was hoping to see more analysis on the Mississippi River Basin dataset and comparisons between it and the DJ dataset where possible. The manuscript seems to be split between two separate analyses but the abstract and introduction do not suggest that. I recommend the author to modify these initial sections as necessary and compare the MRB data with the DJ data where possible. What kind of relation does the author obtain between bankfull depth and bankfull discharge for the MRB under the geotechnical considerations? On the other hand, could clay contents (and cohesion) be estimated with a revised version of equation (4) for other rivers in the world where soil data is not readily available?

**3 Specific comments:**

I list a few specific comments here. Some relate to clarification, others to typos and the last one is a personal opinion, which the author is free to disregard.

1. How do the bankfull estimates found here for the MRB compare to those of Dong et al (2019). This reference appears in the introduction but is not mentioned in the discussion. 2. I did not understand the fourth criteria used to keep a USGS gaging station in the analysis of the MRB. 3. If the analysis discards rivers with depths smaller than 2m, why is the 0.5m to 1.5m soil depth the only section considered for the analysis. What about river sections with different bank layers? The author mentions that soil data below 1.5m is not reliable but how valid is it to assume a uniform soil profile for the entire channel depth? How sensitive is the proposed model to this assumption? 4. Line 150(and other locations) - Ohata et al 2017 (not 2019). 5. Line 151 – How did the author "cross-reference" the Ohata et al. dataset with the Dunne and Jerolmack dataset? 6. Line 214 – what is the equation of the curve (envelope) used to identify the conditions conducive to dune/ripple development? 7. Line 259 (and other locations) – Chen 1969 (not 1971) 8. Line 286 – Vegetation. What are the predominant types of vegetation along the MRB? How deep are their roots? Root length might set slump block thickness. Vegetation might be the most relevant factor in shallow channels (up to max root length) and geotechnical considerations might be more relevant in deeper channels where roots might not stabilize the full bank. 9. I think that box plots might look better than the grey areas used by the author to summarize results within certain bins in the different plots.
**4 References:**

1. Akayuli C., Ofosu, B., Nyako, S.O., and Opuni, K.O. The Influence of Observed Clay Content on Shear Strength and Compressibility of Residual Sandy Soils. International Journal of Engineering Research and Applications (IJERA) ISSN: 2248-9622 www.ijera.com Vol. 3, Issue 4, Jul-Aug 2013, pp.2538-2542, 2013.

2. ASCE Task Committee on Hydraulics, Bank Mechanics, and Modeling of River Width Adjustment: River width adjustment. 1. Processes and mechanisms, Journal of Hydraulic Engineering, 124(9), 881–902, 1998.

3. Chen, W. F., Giger, M. W., and Fang, H. Y.: On the limit analysis of stability of slopes, Soils and Foundations, 9(4), 23–32, https://doi.org/10.3208/sandf1960.9.4_23, 1969.

4. Cisneros, J., Best, J., van Dijk, T. et al. Dunes in the world's big rivers are characterized by low-angle lee-side slopes and a complex shape. Nat. Geosci. 13, 156–162 https://doi.org/10.1038/s41561-019-0511-7, 2020.

5. Dafalla, M. A.: Effects of clay and moisture content on direct shear tests for clay-sand mixtures, Advances in Materials Science and Engineering, 562726, https://doi.org/10.1155/2013/562726, 2013.

6. Dong, T. Y., Nittrouer, J. A., Czapiga, M. J., Ma, H., McElroy, B., Ilicheva, E., et al.: Roles of bank material in setting bankfull hydraulic geometry as informed by the Selenga River Delta, Russia. Water Resources Research, 5, https://doi.org/10.1029/2017WR021985, 2019.

7. Dunne, K. B. J., and Jerolmack, D. J.: Evidence of, and a proposed explanation for, bimodal transport states in alluvial rivers, Earth Surface Dynamics, 6, 583–594, https://doi.org/10.5194/esurf-6-583-2018, 2018.

8. Langendoen, E., Simon, A., and Thomas, R.E. CONCEPTS – A Process-Based Modeling Tool to Evaluate Stream-Corridor Restoration Design. Wetlands Engineering River Restoration Conference, August 27-31, 2001.

9. Masada, T. Shear Strength of Clay and Silt Embankments. Ohio Research Institute

for Transportation and the Environment. Report No. FHWA/OH-2009/7, 319pp. 2009.

10. Motta, D., Abad, J.D., Langeondoen, E., and Garcia, M.H. A simplified 2D model for meander migration with physically-based bank evolution. Geomorphology. Volumes 163–164, Pages 10-25, 2012.

11. Ohata, K., Naruse, H., Yokokawa, M., and Viparelli, E.: New bedform phase diagrams and discriminant functions for formative conditions of bedforms in open-channel flows, Journal of Geophysical Research Earth Surface, 122, 2139–2158. https://doi.org/10.1002/2017JF004290, 2017.

12. Terzaghi, K., Peck, R.P., and Mesri, G. Soil Mechanics in Engineering Practice. Third edition. John Wiley and Sons, Inc. United States of America. 664pp. 1996.

---

## Referee Comment (RC3) · Christopher Hackney (Referee) · 24 Aug 2020

In this manuscript the author sets out to demonstrate that bankfull heights of river banks, and thus the well established hydraulic geometry equations, can be defined by cohesion modulated geotechnical stability. The author tests their hypothesis for a large (387) database of river banks in the Mississippi River Basin (MRB). The author uses large scale soil databases to derive geotechnical properties of the river banks, a welcome approach to address data paucity. The results show that river bank height does indeed appear to be controlled by clay content and the ensuant bank cohesion

defining geotechnical stability. This manuscript comprises a large dataset and the author has done a thorough job of compiling and testing the data, and associated errors implicit in the soil survey data used. I think that the application of such a diverse dataset to this problem warrants publication, but I have a few comments and queries regarding some of the assumptions and discussions made in the paper, detailed below, which I hope the author finds useful.

The proposition of the manuscript (ln 9) is that bankfull depth is predominately defined by the maximum geotechnical stable height of the banks for a given cohesion. Once the critical bank height is exceeded, failure will occur resulting in a self-limiting mechanism on bankfull depth. By definition, if a bank exceeds its stable height for a period of time, it will fail back to a stable height. In the strictest sense, this proposition is not new or previously unexplored. Indeed, as stated by the author in line 34 "This paper demonstrates that bankfull depths predicted by a bank-stability model correlate with observed bankfull depths estimated….in the Mississippi River Basin". The bank-stability model used is that of Chen et al (1971) and the ASCE (1999) – lines 61 – 75 and Eq's 2 -4. Thus, the model in itself is not new, so in effect what this paper is doing is applying a well-established model of critical bank height to a large set of observed data to demonstrate that the model matches observed bank heights. Is this, therefore, a novel premise if the model is known to represent the physical well and has been widely applied? By definition, a channel can not be deeper than the distance between the maximum stable bank height and the deepest part of the channel (for a persistent period for time), therefore it is likely that global (or large) datasets will average out at this maximum stable height as a statistical average.

I find the section between lines 30 and 34 quite confusing. The author states that channel incision or floodplain deposition may increase bank height (OK so far). This causes banks to collapse once a critical bank height is reached. The subsequent failure results in channel widening which tends to increase water depths back towards the stable bank height. It is this last bit that I find counter to the previous few lines. I

follow the argument that a wider channel results in slower flows, but the reduction in flows is a result of the increased channel capacity due to increasing width, and so there is no need for the channel to continue to incise of raise water depths to hold the same volume of water, as the increased width accounts for this. Furthermore, it has been shown recently that over long time frames, channels tend to maintain an equilibrium width (and thus presumably bank height; see Mason and Mohrig, 2019) and that the channel adjustment due to bank collapse also often sees increased deposition on the inner bank (see van de Lageweg et al. 2014). A clearer explaination of how bank failures can result in increased water depths would be welcome.

The author also notes a few potential limitations to the data exploration in the discussion, but does so briefly and in passing. I would like to see a more developed discussion around the role of vegetation induced cohesion, and also the role of failed material (particularly in clay rich soils which are more likely to fail and persist in blocks at the base of the river bank), as these are likely to be key local controls on any variation in the relationships the author has presented. Another potential source of variaiton that isn't raised but may also be important is the role of floodplain topography in defining bankful depths and bank heights. If a bank is eroding through a scroll-bar then it is likely that following a failure the local bank height may decrease as a result of variable local topography (i.e. on a floodplain where elevation is sloping away from the bank). Following the model presented here, will that new bank remain stable until its critical height either is reached through channel incision or build up from floodplain deposition?

Following on from this, on Ln 272 the author states that an increase in bank height caused by floodplain deposition may trigger bank failure. Presumably, to deposit material on the bank the flow needs to be over-bank. Therefore, is it the deposition of the material on the bank during these flows, for the increased water velocity and bank shear stresses that will induce this erosion?

Overall, I think a lot of the issues raised above come down to the temporal scale being examoined here and there is a need for some discussion the manuscript around the

time-scales over which these processes may become the dominant factor and how time-averaging of the other processes involved in river bank erosion occurs. Could the author examine any historical rates of bank erosion for sites analyses in the manuscript to see whether the theoretical model holds for different time periods?

A small point on the colouring of figure 1: To my eyes the background panel surround the basin appears the same as the 30% clay content colour and is very confusing. Could this be changed to avoid confusion.

Refs not included in original manuscript: Mason, J. and Mohrig, D. Differential bank migration and the maintenance of channel width in meandering river bends, Geology (2019) 47 (12): 1136–1140.

van de Lageweg, W.I. and van Dijk, W.M. and Baar, A.W. and Rutten, J. and Kleinhans, M.G. (2014) 'Bank pull or bar push : what drives scroll-bar formation in meandering river?', Geology., 42 (4). pp. 319-322.

---

## Author Comment (AC1) · 28 Oct 2020

I wish to thank the reviewers for their insightful comments, which have led to a significantly improved paper (which I will submit if the editor deems my responses below to be sufficient for the review process to continue). I apologize that it has taken so long to provide responses. Here I summarize the points of each reviewer, respond to each point, and state how the revised manuscript has been or will be modified to address each comment.

[Figure]

Reviewer 1 (Gordon Grant):

Q: "I remain somewhat skeptical of both the approach and the results. My skepticism is rooted in both some of the underlying theory and assumptions used to derive the mathematical relations that underpin this work, and also the wide variance and assumptions in the data used to validate the methods. I cannot put my finger on a single "smoking gun" amidst all of this, but I was quite surprised that given all the assumptions and uncertainties, the predicted relationships were almost spot-on the widely published values (starting with Leopold and Maddock) for the hydraulic geom-etry relationships. Herein lies the intrigue, but I'm not convinced that the right answer was obtained for the right reasons. To his credit, the author acknowledges the fragility of some of the assumptions, and calls for additional work to better understand key relationships. Thus, this paper will stimulate discussion and work and deserves to be shared with the broader community, irrespective of any misgivings."

A: I appreciate that the reviewer sees value in the paper despite his skepticism.

Q: "The postulated relation between bankfull height and clay content builds on previous work, but as the author points out, the assumption in this earlier work was that cohesion limited erosion by increasing the required near-bank shear stresses for fluvial entrainment. Here the focus is on gravitationally-driven stress failures, modeled with a very simple 1-D relationship (Eq. 2), that when parametrized, gives rise to an even simpler linear equation between bankfull depth and maximum gravitationally stable bank height. This relation is tested at the scale of the Mississippi River basin by comparing bankfull height derived from USGS gage station data with predicted height calculated from clay content derived for each station using gNATSGO soil survey data and estimated with a 10km moving window. Not surprisingly, the resulting scatter plot is. . .well, scattered with a weak positive trend (Fig. 2A). The author does synthetically consider what the error in this relationship might look like; this error analysis is not propagated hrough the rest of the analysis, however. At the end of the day, I remain skeptical that broad-scale soil survey data can be used to parametrize clay content for

point data (USGS stations). To be fair, in the discussion the author does consider other sources of potential error beyond data error, including the use of stage data as a proxy for bank height, and other possible channel adjustments to varying clay content (i.e., varying bank angle, widening). But taken together, these uncertainties raise doubts about the validity of the paper's central claim that bankfull height is primarily controlled by gravi-tational slope failure."

A: Thank you for this comment. First, let me state that my discussion paper did not claim that gravitational failure is the dominant process occurring in channel widening and bank migration more generally. Fluvial scour and gravitational failure influence and complement one another to such an extent that it is difficult to clearly identify a dominant control for even a single alluvial channel, let alone for alluvial channels in general. Fluvial scour transports slumped material away from the bank, likely keeping bank angles higher than they would be without fluvial scour. Gravitational failure transports material from higher on the bank to the toe (often following fluvial scour, which is maximized at the toe), likely keeping bank angles lower than they would be without gravitational failure. The relative importance of these two processes may be reflected in the bank angle, i.e., a bank angle persistently much less than vertical may indicate the dominance of gravitational failure while a bank angle persistently at or above 90ËŽ (i.e., a cantilever or overhang) may indicate the dominance of fluvial scour.

Given that both gravitational failure and fluvial scour play important and complementary roles, which process do data suggest is the more dominant control on alluvial channel morphology? This question can be addressed by comparing the channel geometries predicted by models based on gravitational failure and fluvial scour to data. Dunne and Jerolmack (2020) tested the fluvial scour hypothesis using the same alluvial channel morphology dataset (i.e., Dunne and Jerolmack, 2018) used in my paper. The authors explain why their model underpredicts data by approximately 2 orders of magnitude in channels with D50 < 1 cm (Fig. 2A of their 2020 paper) this way: "For fine-grained rivers with D50 < 1 cm, however, we see rivers peel off of the Shields curve;

[Figure]

the smaller the riverbed grain size, the larger bankfull shear stress deviates from the threshold expectation... we infer that this departure... represents the point where c of cohesive banks, which is rarely measured, becomes larger than c of noncohesive bed sediments, on average." The predictions of a model based on gravitational failure, while scattered, are consistent with data for all grain sizes. The only systematic deviation of a model based on gravitational failure occurs for shallow channels (< 2 m deep) with very low clay contents, a result that, as noted in the discussion paper, may be a result of an upward bias in clay content estimates at very low clay contents. I believe such a bias exists because low clay contents can only be constrained within the range from 0 to 10% clay, the average value of which (5%) is often assigned to the soil horizon being measured. If the actual value is very close to 0 (e.g., < 1%), the clay content estimate will be overestimated by much more than a relatively large clay content will be underestimated (e.g., 5% clay is 400% greater than 1% clay but only 50% lower than 10% clay). The averaging of clay contents from different horizons within the same soil and in different soils within a floodplain adjacent to a channel may partly mitigate this problem, but it is difficult to estimate low clay contents in the field in any case. I excluded channels with clay contents close to zero from my analysis due to this potential bias (line 205).

I did not propagate uncertainties more comprehensively in my discussion paper because there are several sources of uncertainty, not all of which are straightforward to quantify: uncertainty in measured clay content, uncertainty in the relationship between cohesion and clay content, and uncertainty in the bank angle and moisture content that determine the stability parameter Ns. For example, as lines 257-259 noted, variations in bank angle can result in Ns values that differ by a factor of 10, so working to find additional constraints on bank angles and how they vary throughout the MRB might quickly make obsolete the estimation and propagation of uncertainties in the current analysis, which implicitly assumes uniform Ns values. That said, I agree with the reviewer that a more comprehensive approach to uncertainty propagation would strengthen the paper. I am still working on how best to quantify and propagate uncertainties in the revised

manuscript, but the steps I am attempting include: 1) estimating and propagating the uncertainty in clay content within gNATSGO to be approximately 10% (absolute value) based on similar datasets (e.g., Ramcharan et al., 2018), 2) quantifying and propagating the uncertainty in the least-squares linear regression between cohesion and clay content using the data from Dafalla (2013), and 3) compiling data on bank angles from the MRB (where available) to estimate uncertainty in bank angle and the related uncertainty of Ns values.

I appreciate the reviewer's skepticism that broad-scale soil survey data can be used to parameterize clay content for point data. However, I note that soil survey data are derived from point-scale measurements (SCS scientists use data from soil pits to define soil properties across areas of similar slope, geology, etc.). Moreover, rates of bank retreat at a gaging station are likely controlled not by the clay content of a single point but rather by clay content over the spatial scale of a meander bend, which for large rivers can be ∼1-10 km.

Q: "While I'm appreciative of the author's invoking the critical flow hypothesis (correct citation is Grant, 1997, not 2000 as it appears in text but not references), I'm somewhat confused by the role it plays in this story. As I understand it, the author argues that this hypothesis suggests that the range of Froude numbers for both sand and gravel-bed channels should be limited to near- or less than 1.0, and that therefore Froude number and discharge should be weakly and inversely related. The logic here is not entirely clear, and the mechanism that restricts Froude numbers is not entirely accurate. In its original form, the hypothesis argues that in steep channels (typically S>0.01), interactions between the free surface (particularly hydraulic jumps) and the bed result in a rough balance between forces that accelerate the flow and forces that extract energy from the flow and thus retard it, thus promoting near-critical flow conditions. This condition applies irrespective of grain size, although the author rightly points out that the actual bed features that set up this interaction are different for sand, gravel, and even boulder bed channels. A recent paper tests this idea in the flume and articulates the

mechanism well (Piton and Recking, 2019). Most of the streams in the Dunne and Jerolmack data set have slopes much less than 0.01, and consequently much lower Fr, as the data shown in the paper point out."

A: I thank the reviewer for pointing out the incorrect citation, which has been corrected in the revised manuscript. The reviewer is correct that I generalized his 1997 critical flow hypothesis. The wording has been corrected in the revised paper. My reason for generalizing his hypothesis was that, even though the critical flow hypothesis was originally proposed for steep channels, the key idea (to my mind) of the critical flow hypothesis is the existence of a self-regulatory feedback in which an increase in velocity is met with an increase in drag that brings the velocity, and hence Fr, back down to a lower value. Similarly, a decrease in velocity tends to be met with a decrease in drag that tends to increase the velocity. It is likely that such a self-regulatory feedback is not limited to steep channels. In steep, gravel-bedded channels, faster flow tends to lead to more wave breaking and other drag-inducing processes, thereby tending to lower velocities back down to a critical value (i.e., Fr = 1). In less-steep, sand-bedded channels, faster flow tends to lead to the development of larger and more well-developed bedforms, which increases relative roughness and hence drag in the range of Fr values associated with well-developed bedforms (Fr $\sim$ 0.1-1). The mechanisms for self-regulation differ between steep channels and less-steep channels, but self-regulation exists in both. In the revised manuscript, I clarify that the critical flow hypothesis applies to steep channels only and separately discuss evidence for a similar self-regulation of Fr in less-steep channels.

Q: "More to the point, in my view, the primary control on Fr is channel slope, and I suspect that this is what is behind the weak inverse correlation between Fr and discharge (Fig. 3A). The simple theoretical dependency between Fr and S is shown in Grant (1997; ig. 4); a more sophisticated treatment is given in Palucis and Lamb (2017; Fig. 2). Since smaller streams tend to be steeper than larger ones, my hunch is that discharge is more of a proxy for slope than a driver as in Fig. 3A. This slope dependency
is also probably lurking behind the grain-size/Fr relationship in Fig. 3B as well. None of this fundamentally invalidates the argument being made by the author but I think these relations should be acknowledged, as they have bearing on the physical mechanisms underlying the presentation."

A: Thank you for this comment. My work demonstrates how alluvial channel geometric variables interrelate. I used bankfull discharge as an independent variable because it has traditionally been used for this purpose. I did not and would not argue that any variable is more fundamental, significant, or a "driver" than any other variable. Bankfull Froude number depends explicitly on channel depth and velocity, the latter of which depends on slope and bed roughness. As such, bankfull Froude number certainly depends on channel depth, channel slope, and bed roughness (the latter of which depends on grain size and all of the factors that control bedform geometry (if bedforms are present)). I know of no basis for ranking the relative importance of any of these controls. This view is consistent with Palucis and Lamb (2017), who, despite relating specific channel morphologic types to specific slopes, document the importance of channel width and grain size in controlling channel type (e.g., "step‐pools form in near supercritical flow or when channel width is narrow compared to bed grain size") and conclude that "certain bed slopes have unique channel morphologies because the process variables covary systematically with bed slope."

Reviewer 2 (Roberto Fernandez):

Q: "1. The use of the data by Dafalla (2013) seems misrepresented in this manuscript (Figure S1). Cohesion values reported by Dafalla (2013) correspond to pure sand, sand and clay mixtures with 5 In addition, Dafalla (2013) shows that for the same clay content (15 I really believe that the approach proposed in this manuscript has a lot of potential and others have included geotechnical considerations in models for stream restoration (e.g. CONCEPTS, see Langendoen et al. 2001; RVR Meander, see Motta et al. 2012). I would encourage the author to dig out some more references regarding cohesion estimates for soils that are more relevant for the Mississippi River

Basin. For example, Masada (2009) presents a very extensive report on geotechnical parameters for the state of Ohio and includes different relations between sediment properties (cohesion for example) and soil composition (amount of silt, amount of clay, etc.). Their approach is specific to highway embankments but the results of their tests might be more general in terms of cohesion values in relation to clay contents."

A: I greatly appreciate the reviewer's drawing my attention to these additional sources of data, Masada (2009) in particular. I have scoured the literature for additional sources of data but have not found any. Masada (2009) is not a good source of data on the relationship between cohesion and clay content because he presents cohesion versus clay content for a very narrow range of clay contents (e.g., 27%-34% in Fig. E6). Also, I prefer to use data from direct shear tests rather than the triaxial shear tests of Masada (2009) because direct shear tests are more similar to gravitational shear failures. Dafalla (2013) is also a good source of data because he presents data for cohesion versus clay content at fixed moisture contents. Assuming that there is no systematic variation between short and tall banks in the water content at which channel banks tend to fail, it is the dependence of cohesion on clay content at similar moisture contents (or averaged across all moisture contents) that is most appropriate for use in the model.

The reviewer is correct that the sand and clay mixtures considered by Dafalla (2013) were obtained by mixing sand and clay as opposed to being natural soils. However, I don't think creating such mixtures necessarily undermines the quality of the data or the applicability of the results to natural soils. Masada (2009) considered natural soils but they were almost entirely glacial in origin, hence not necessarily a better analog for alluvial soils than the mixtures studied by Dafalla (2013).

Q: "2. The use of equation (2) might not be appropriate for riverbanks. The use of that equation as presented by Chen (1969) and Terzaghi et al. (1996; p. 271-272) is for soil embankments located above the water table. Several authors have used it in the past as discussed by ASCE (1988) but even there, the authors suggest that critical depth

approaches are not accurate when the most common bank failure mechanisms for riverbanks are due to tension cracks that cause toppling or cantilever failures. Assuming the equation is indeed an appropriate approach for riverbanks, I would en-courage the author to explore the sensitivity of its input variables to other values. Chen (1969) shows a wide range of Ns values that depend on the internal friction angle of the material (which is sensitive to moisture content) and the actual slope of the bank. The smallest stable bank height would be given by the smallest possible safety param-eter Ns so why not explore a range of Ns values. When the channel has low flow, the bank might be quite dry and its maximum stable height would be quite different from that obtained with a saturated bank (e.g. during the falling limb of a hydrograph where the river stage is getting lower but the bank remains saturated). It would be very useful to see these considerations in the analysis. The author discusses the issue briefly but more details regarding bank failure mechanisms and their prevalence might strengthen the manuscript."

A: Thank you for this comment. The results of eqn. (2) are consistent with more complex models if Ns values are defined in such a way as to incorporate the effects of moisture content. A linear relation between the maximum stable bank height and cohesion is robust across multiple bank-stability models with a variety of water table or moisture content scenarios. For example, Eaton (2006) derived a linear relationship between maximum stable bank heights and cohesion for fully saturated banks. Simon and Thomas (2002) demonstrated how the maximum stable bank height decreases with both an increasing bank angle and an increasing relative height of the water table. These results are consistent with Chen et al. (1969) if Ns values incorporate the effects of moisture content and/or water table height.

In the revised manuscript I will provide a sensitivity analysis that demonstrates the effects of variability and uncertainty in Ns values on the model predictions, as the reviewer suggests. I agree with the reviewer that channel banks may be quite stable when dry. However, over long time scales banks are saturated or near-saturated many

times, hence it is the shear strength during these times of relative weakness that are most relevant to determining the long-term stability of a bank.

Q: "3. Sensitivity analysis: Figures 2b and 2c present results for bank heights based on a synthetic dataset. If the author estimated clay contents using averaging windows for a soils dataset, why not extract second order statistics from it and use them directly instead of creating a synthetic dataset?"

A: Thank you for this comment. However, I don't think that second-order statistics (e.g., coefficient of variation) necessarily capture all of the potential uncertainty in a dataset. For example, spatial variations in the values of a dataset that systematically over- or under-predict actual values won't correctly capture the true uncertainty. I also think creating a synthetic dataset has advantages. For example, a key goal was to demonstrate that an upward bias can exist in very low clay contents. I don't think using second-order statistics in the averaging would necessarily have the same potential to demonstrate this phenomenon.

Q: "4. Use of the Mississippi River Basin data: The author clearly states why the MRB data are used. However, not knowing much about the many different locations along the basin, I have a few questions. (1) What percentage of the cross sections analyzed can be considered natural? (2) Did the author discard those locations where the navigable channels are maintained by the US Army Corps of Engineers? (3) Of the many stations used, how many might be influenced by river control structures (dams, wing dams, chevrons, etc.) or road infrastructure (e.g. culverts, bridges)?"

A: Thank you for this comment. Using Google Earth, I have examined the locations of the 354 stations and have found no instances where cross-sections are located close to infrastructure. This lack of overlap between the U.S.G.S. station locations and infrastructure may be partly due to the fact that my filtering criteria (lines 88-92) did an effective job at removing stations with stage-discharge relationships that are affected by infrastructure.

Q: "5. Figures 3 and 4: It is not at all clear why the author includes regression plots of the Dunne and Jerolmack (2018) dataset. Based on the abstract and introduction, it was unexpected that a different dataset appears in the manuscript and becomes the focus of the second half. I understand the use of the dataset for Figure 5, which is new but the content of Figures 3 and 4, is not. I would encourage the author to make it clear to the reader earlier that the DJ dataset is a substantial part of the analysis and to state explicitly the novelty of including figures 3 and 4."

A: Thank you for this comment. The opening sentence of the abstract makes clear that the problem I am tackling is the power-law scaling of bankfull depths, widths, depth-averaged water velocities, and along-channel slopes to bankfull discharge in alluvial channels. The first part of my paper deals with control on channel depth only, so it stands to reason that there must be another part of the paper that extends the work on channel depth to other aspects of channel geometry using additional principles. Figures 3 and 4 (including the reporting of best-fit exponents) have not been published elsewhere and their inclusion is important for meeting the goals of the paper.

Q: "6. Figure 5: I have a few specific questions about the analysis leading to Fig. 5. (1) What is the number (and percentage) of cases that report ripples/dunes over the entire Ohata (2017) dataset? (2) For those reporting ripples/dunes, what is the number and percentage of measurements obtained in the laboratory and in the field? (3) For those in the field, how many are for large rivers? Cisneros et al. (2020) show that traditional dune scaling equations overestimate the size of dunes in large rivers and propose the following relation between dune height (H) and water depth (h) – H 0.056h - 0.12h. 4) Are the only sources of roughness in the DJ data the ripples/dunes or gravel size? What about bars, meandering, vegetation?"

A: Thank you for this comment. 1) 1574 (42%) of the 3790 data points in Ohata et al. (2019) have ripples or dunes (noted in revised manuscript), 2) of that 42%, 19% are in the field and the remaining 81% are in the laboratory (noted in revised manuscript), 3) 3.2% (123 out of 3791) of the data points are from rivers with h > 5 m. 4) I am assuming that the dominant (not only) sources of roughness in the channels of the DJ data are ripples/dunes or gravel clasts. Long-wavelength topographic features such as bars and meanders are not likely to be dominant roughness/drag-inducing elements given that the presence/absence of the flow separation that tends to dominate drag depends sensitively on the maximum slope of bedforms and other obstacles to the flow, with slopes in excess of 0.2 m/m generally needed to trigger the flow separation (though surface curvature also plays an important role in addition to slope; see below). I concede that vegetation can be a dominant source of roughness on the beds of ephemeral channels, and some of the scatter in my analysis may be a result of vegetation-induced bed roughness.

Cisneros et al. (2020) demonstrates that the lee-side angle of many bedforms in large alluvial channels is lower that empirical equations predict and argues that such lower angles means that flow separation and hence drag is less significant than empirical models would suggest in large rivers. It is important to note, however, that flow separation depends sensitively on the surface curvature in the zone of the maximum adverse pressure gradient (Lamballais et al., 2010), not just on the relative height or lee-side angle of bedforms. As such, more research is needed to conclude that flow separation is rare on the lee sides of bedforms in large alluvial channels.

Q: "As a final general comment, I was hoping to see more analysis on the Mississippi River Basin dataset and comparisons between it and the DJ dataset where possible. The manuscript seems to be split between two separate analyses but the abstract and introduction do not suggest that. I recommend the author to modify these initial sections as necessary and compare the MRB data with the DJ data where possible. What kind of relation does the author obtain between bankfull depth and bankfull discharge for the MRB under the geotechnical considerations? On the other hand, could clay contents (and cohesion) be estimated with a revised version of equation (4) for other rivers in the world where soil data is not readily available?"

A: Thank you for this comment. See my response to point 5. The revised manuscript

will include the relation between bankfull depth and bankfull discharge for the MRB under the geotechnical considerations. It should be possible to infer clay content values for some rivers if bank angles and moisture contents were also well constrained. We do not have such data yet for many alluvial channel cross sections across a sizable region.

Q: "1) How do the bankfull estimates found here for the MRB compare to those of Dong et al (2019). This reference appears in the introduction but is not mentioned in the discussion. A: Since Dong et al. (2019) deal with the Selenga River Delta and my paper deals with the MRB, it is difficult to make a direct comparison with their results.

Q: "2. I did not understand the fourth criteria used to keep a USGS gaging station in the analysis of the MRB."

A: I apologize that this was not clearer. The estimation of bankfull stage requires fitting stage to discharge for the smallest and largest values. The bankfull stage is estimated to be where these two lines meet. In order to verify that the low-flow fit is reasonable, I retained only those stations for which the extrapolation of the fit of low-flow values passes "close" to the correct value: zero flow depth at zero discharge. In cases where the extrapolation of the fit does not pass close to zero, the data are likely of insufficient quality or require an adjustment based on data that are not publicly available. What represents "close" should not be based on an absolute error, e.g., 0.5 m, because such a criterion would require that low-flow fits for a deep channel be much more accurate than one for a shallow channel. So, instead, I defined "close" as within 50% of the bankfull stage from zero. That is, if the bankfull stage is 5 m, then the extrapolation of the low-flow fit to zero discharge must yield a stage within 2.5 m of zero. Similarly, if the bankfull stage is 2 m, then the extrapolation of the low-flow fit to zero discharge must be within 1 m of zero. An abbreviated version of this explanation is included in the revised manuscript.

Q: "3. If the analysis discards rivers with depths smaller than 2m, why is the 0.5m to

1.5m soil depth the only section considered for the analysis. What about river sections with different bank layers? The author mentions that soil data below 1.5m is not reliable but how valid is it to assume a uniform soil profile for the entire channel depth? How sensitive is the proposed model to this assumption?"

A: Thank you for this comment. I assume that the texture of the top 1.5 m of the bank (for which data is readily available) is representative of the entire bank. I understand that this assumption may be violated in many cases. However, given that the floodplain deposits that comprise many banks are the depositional products of channels where well-sorted sediments tend to be the norm, it is reasonable to expect that the texture of the uppermost 1.5 m of the bank will correlate strongly with bank material at greater depths in many cases.

Q: "4. Line 150(and other locations) - Ohata et al 2017 (not 2019)." A: That you for pointing out this typo. It has been fixed in the revision.

Q: "5. Line 151 "How did the author "cross-reference" the Ohata et al. dataset with the Dunne and Jerolmack dataset?

A: Thank you for this comment. Cross-referencing refers to the development of a curve in Fr vs. d50 space that separates channels that have ripples and/or dunes from those that do not, and using that curve to infer that the vast majority of sand-bedded channels in the Dunne and Jerolmack dataset have ripples and dunes. This is done by assuming the existence of ripples and dunes in channels of the D&J dataset that have Fr and d50 values that sit above and to the left (Fig. 3B) of the envelope curve separating channels with and without ripples and dunes in the Ohata et al. (2019) dataset. This has been clarified in the revision.

Q: "6. Line 214 "what is the equation of the curve (envelope) used to identify the conditions conducive to dune/ripple development?"

A: There is no equation. This is simply a drawn curve. Noted in revised manuscript.

Q: "7. Line 259 (and other locations) "Chen 1969 (not 1971)""

A: Typo fixed.

Q: 8. Line 286 "Vegetation. What are the predominant types of vegetation along the MRB? How deep are their roots? Root length might set slump block thickness. Vegetation might be the most relevant factor in shallow channels (up to max root length) and geotechnical considerations might be more relevant in deeper channels where roots might not stabilize the full bank.

A: Thank you for this comment. The stability of any bank is primarily a function of its weakest portion or layer, i.e., failure is more likely to occur in a zone of low shear strength compared to a zone of high shear strength, all else being equal. Failure of a weaker zone underlying a stronger zone can create a cantilever, but such cantilevers cannot continue to grow indefinitely, hence rates of long-term retreat in stronger and weaker zones of the same bank will tend to be similar. This, together with the fact that vegetation is likely to strengthen just the uppermost approximately 2 m of channel banks (globally, >99% of roots are found in the top 2 m of the soil (Jackson et al., 1996)), suggests that bank material shear strength is not likely to be controlled primarily by vegetation in channels deeper than 2 m. I concede that some plant roots can exceed (even greatly exceed) 2 m in depth. However, plant roots become quite small in density at > 2 m depth based on Jackson et al. (1996). I understand that this view appears to contradict the dependence of bank retreat rates on vegetation that has been documented in many studies (e.g., Ielpi and Lapôtre, 2020). I am not stating that vegetation has no effect on bank stability. Rather, I am stating that I know of no study that has attributed bank shear strength to vegetation that has accounted for the fact that wetter climates with more vegetation also tend to have more clay-rich soils. As such, much of the apparent control of vegetation could be due to clay content.

Reviewer 3 (Christopher Hackney):

Q: "I find the section between lines 30 and 34 quite confusing. The author states

that channel incision or floodplain deposition may increase bank height (OK so far). This causes banks to collapse once a critical bank height is reached. The subsequent failure results in channel widening which tends to increase water depths back towards the stable bank height. It is this last bit that I find counter to the previous few lines. I follow the argument that a wider channel results in slower flows, but the reduction in flows is a result of the increased channel capacity due to increasing width, and so there is no need for the channel to continue to incise of raise water depths to hold the same volume of water, as the increased width accounts for this. Furthermore, it has been shown recently that over long time frames, channels tend to maintain an equilibrium width (and thus presumably bank height; see Mason and Mohrig, 2019) and that the channel adjustment due to bank collapse also often sees increased deposition on the inner bank (see van de Lageweg et al. 2014). A clearer explaination of how bank failures can result in increased water depths would be welcome."

A: Thank you for this comment. The reviewer is quite correct that the sentence "Bank failure results in channel widening, which may reduce depth-averaged water velocities and therefore tend to increase water depths (to convey similar water discharges) back towards the maximum stable bank height" was confusing and did not accurately convey my conceptual model for how alluvial channels self-regulate to a maximum depth comparable to the maximum stable bank height. I have replaced it with: "Bank failure results in channel widening, which introduces new sediment into the channel bed. As this sediment is redistributed on the bed, the channel depth tends to decrease back towards its critical value."

Q: "The author also notes a few potential limitations to the data exploration in the discus-sion, but does so briefly and in passing. I would like to see a more developed discus-sion around the role of vegetation induced cohesion, and also the role of failed material (particularly in clay rich soils which are more likely to fail and persist in blocks at the base of the river bank), as these are likely to be key local controls on any variation in the relationships the author has presented. Another potential source of

variaiton that isn't raised but may also be important is the role of floodplain topography in defining bankful depths and bank heights. If a bank is eroding through a scroll-bar then it is likely that following a failure the local bank height may decrease as a result of variable local topography (i.e. on a floodplain where elevation is sloping away from the bank). Following the model presented here, will that new bank remain stable until its critical height either is reached through channel incision or build up from floodplain deposition? Following on from this, on Ln 272 the author states that an increase in bank height caused by floodplain deposition may trigger bank failure. Presumably, to deposit ma-terial on the bank the flow needs to be over-bank. Therefore, is it the deposition of the material on the bank during these flows, for the increased water velocity and bank shear stresses that will induce this erosion?"

A: Thank you for this comment. Discussion items are, by nature, relatively brief (in that they are limited to subsections of the Discussion section only). To the extent that my discussion items are relatively brief, I think this is appropriate given that there is no indication that vegetation-induced cohesion causes the scaling of alluvial channel geometry that is the subject of this paper. Also please see my response to reviewer 2's comment 8 on the issue of vegetation.

The revised manuscript mentions the role of failed bank material in the context of the interplay between fluvial scour and gravitational failure in contributing to channel width adjustment.

Regarding the final question, either incision or floodplain deposition can trigger bank failure: it is not one or the other. Whether a channel incises depends on changes in base level and sediment supply that depend on tectonic and climatic processes that operate at scales much larger than a channel reach. A channel that is relatively shallow relative to its bankfull depth will tend to flood overbank more often than a deeper channel, thereby promoting deposition until the channel deepens to the point where channel banks fail. The model of this paper invokes a long-term dynamic equilibrium between channel deepening processes (channel incision and/or floodplain deposition)

and channel shallowing processes (bank retreat, which introduces sediment into that channel that, once redistributed laterally, causes channel aggradation). My response to the reviewer's first concern clarifies this aspect of the conceptual model in the revised manuscript.

Q: "Overall, I think a lot of the issues raised above come down to the temporal scale being examoined here and there is a need for some discussion the manuscript around the ime-scales over which these processes may become the dominant factor and how time-averaging of the other processes involved in river bank erosion occurs. Could the author examine any historical rates of bank erosion for sites analyses in the manuscript to see whether the theoretical model holds for different time periods?"

A: I appreciate this suggestion. However, while rates of bank erosion are available over multiple time scales, what matters to this paper is channel widening. Channel widening likely occurs most abruptly during channel incision periods (following climatic changes, for example), after which bank retreat continues but channel widening does not because cut bank migration is approximately balanced by point-bar progradation (e.g., Mason and Mohrig, 2019). I am not aware of any available data on channel widening data over multiple time scales. I also wish to emphasize, as I did in my responses to reviewer 1, that I am not arguing that gravitational failure is the dominant processes setting channel depth and width. Fluvial scour and gravitational failure influence and complement one another so intimately that it is difficult to clearly identify a dominant control.

References not cited in the discussion paper or reviewer comments:

Dunne, K.B.J., and D.J. Jerolmack (2020). What sets river width? Science Advances, 6(41), eabc1505, doi:10.1126/sciadv.abc1505

Eaton, B.C. (2006). Bank stability analysis for regime models of vegetated gravel bed rivers. Earth Surface Processes and Landforms. 31, 1438-1444.

**ESurfD**
[Figure]

Ielpi, A., and M. Lapotre (2020). A tenfold slowdown in river meander migration driven by plant life. Nature Geoscience, 13, 82-86.

Jackson, R.B., J. Canadell, J.R. Ehleringer, H.A., Mooney, O.E. Sala, and E.D. Schulze (1996). A global analysis of root distributions for terrestrial biomes. Oecologia, 108, 389-411.

Lamballais, E., J. Silvestrini, and S. Laizet (2010). Direct numerical simulation of flow separation behind a rounded leading edge: Study of curvature effects. International Journal of Heat and Fluid Flow, 31, 295-306.

Simon, A., and R.E. Thomas (2002). Processes and forms of an unstable alluvial stream with resistant, cohesive streambeds. Earth Surface Processes and Landforms 27, 699–718.

---

## Author Response (AR1)

I wish to thank the reviewers for their insightful comments, which have led to a significantly improved paper. I apologize for the length of time it has taken me to provide a revised manuscript; my primary focus for the past few months has been on single-parenting during the pandemic and the basics of my professional obligations such as teaching and servicing funded projects. I appreciate everyone's patience and request that if the revised paper falls short in some important way that I be given one more opportunity to strengthen the manuscript. Here I summarize the points of each reviewer, respond to each point, and state how the revised manuscript has been modified to address each comment.

**Reviewer 1 (Gordon Grant):**

Q: "I remain somewhat skeptical of both the approach and the results. My skepticism is rooted in both some of the underlying theory and assumptions used to derive the mathematical relations that underpin this work, and also the wide variance and assumptions in the data used to validate the methods. I cannot put my finger on a single "smoking gun" amidst all of this, but I was quite surprised that given all the assumptions and uncertainties, the predicted relationships were almost spot-on the widely published values (starting with Leopold and Maddock) for the hydraulic geometry relationships. Herein lies the intrigue, but I'm not convinced that the right answer was obtained for the right reasons. To his credit, the author acknowledges the fragility of some of the assumptions, and calls for additional work to better understand key relationships. Thus, this paper will stimulate discussion and work and deserves to be shared with the broader community, irrespective of any misgivings."

A: I appreciate that the reviewer sees value in the paper and wishes to see it published despite his skepticism.

Q: "The postulated relation between bankfull height and clay content builds on previous work, but as the author points out, the assumption in this earlier work was that co-hesion limited erosion by increasing the required near-bank shear stresses for fluvial entrainment. Here the focus is on gravitationally-driven stress failures, modeled with a very simple 1-D relationship (Eq. 2), that when parametrized, gives rise to an even sim-pler linear equation between bankfull depth and maximum gravitationally stable bank height. This relation is tested at the scale of the Mississippi River basin by comparing bankfull height derived from USGS gage station data with predicted height calculated from clay content derived for each station using gNATSGO soil survey data and esti-mated with a 10km moving window. Not surprisingly, the resulting scatter plot is. . .well, scattered with a weak positive trend (Fig. 2A). The author does synthetically consider what the error in this relationship might look like; this error analysis is not propagated hrough the rest of the analysis, however. At the end of the day, I remain skeptical that broad-scale soil survey data can be used to parametrize clay content for point data (USGS stations). To be fair, in the discussion the author does consider other sources of potential error beyond data error, including the use of stage data as a proxy for bank height, and other possible channel adjustments to varying clay content (i.e., varying bank angle, widening). But taken together, these uncertainties raise doubts about the validity of the paper's central claim that bankfull height is primarily controlled by gravitational slope failure."

A: Thank you for this comment. First, let me state that my discussion paper did not claim that gravitational failure is the dominant process occurring in channel widening and bank migration more generally. Fluvial scour and gravitational failure influence and complement one another to such an extent that it is difficult to clearly identify a dominant control for even a single alluvial channel, let alone for alluvial channels in general. Fluvial scour transports slumped material away from the bank, likely keeping bank angles higher than they would be without fluvial scour. Gravitational failure transports material from higher on the bank to the toe (often following fluvial scour, which is maximized at the toe), likely keeping bank angles lower than they would be without gravitational failure. The relative importance of these two processes may be reflected in the bank angle, i.e., a bank angle persistently much less than vertical may indicate the dominance of gravitational failure while a bank angle persistently at or above 90° (i.e., a cantilever or overhang) may indicate the dominance of fluvial scour. The following text has been added to the manuscript on this point:

"That said, gravitational failure and fluvial shear stresses act in concert in such a way that identifying which process is dominant may be difficult. Fluvial scour transports slumped material away from the bank, likely keeping bank angles higher than they would be without fluvial scour. Gravitational failure transports material from higher on the bank to the toe (often following fluvial scour, which is maximized at the toe), likely keeping bank angles lower than they would be without gravitational failure. More research is needed, but the relative importance of these two processes may be reflected in the bank angle, i.e., a bank angle persistently much less than vertical may indicate the dominance of gravitational failure while a bank angle persistently at or above 90° (i.e., a cantilever or overhang) may indicate the dominance of fluvial scour."

Given that both gravitational failure and fluvial scour play important and complementary roles, which process do data suggest is the more dominant control on alluvial channel morphology? This question can be addressed by comparing the channel geometries predicted by models based on gravitational failure and fluvial scour to data. Dunne and Jerolmack (2020) tested the fluvial scour hypothesis using the same alluvial channel morphology dataset (i.e., Dunne and Jerolmack, 2018) used in my paper. The authors explain why their model underpredicts data by approximately 2 orders of magnitude in channels with  $D_{50} < 1$  cm (Fig. 2A of their 2020 paper) this way: "For fine-grained rivers with  $D_{50} < 1$  cm, however, we see rivers peel off of the Shields curve; the smaller the riverbed grain size, the larger bankfull shear stress deviates from the threshold expectation... we infer that this departure... represents the point where  $\tau_c$  of cohesive banks, which is rarely measured, becomes larger than  $\tau_c$  of noncohesive bed sediments, on average."

The predictions of a model based on gravitational failure, while scattered, are consistent with data for all grain sizes. The scatter of the model predictions can be expected to be largely a consequence of lack of data on bank angle.

Text added on this point:

"Absent site-specific data for bank angles, the largest source of uncertainty in the proportionality coefficient in Eq. (4) as applied to specific locations is likely the bank angle, since relatively modest variations in bank angle (e.g., from 90° to 75°) are associated differences in Ns of approximately a factor of 3 while other sources of uncertainty (e.g., between cohesion and clay content as quantified by Eq. (3)) are smaller. Section 4 provides discussion on how uncertainty in bank angle and other factors such as bank vegetation limit the precision of Eq. (4) to specific

locations. The primary of objective of this paper, however, is to document an increase, on average, in bankfull channel depth with increasing clay content:

$$h \approx 0.35 \, p_c.$$
 (5)

assuming that bankfull depth is approximately equal to the maximum gravitationally stable bank height."

The only systematic deviation of a model based on gravitational failure occurs for shallow channels (< 2 m deep) with very low clay contents, a result that, as noted in the discussion paper, is likely a result of an upward bias in clay content estimates at very low clay contents. Text added on this point:

"This upward bias may be associated with the difficulty of measuring very low clay contents in the field. Clay contents estimated in the field can only be constrained to be within the range from 0 to 10% clay. If the actual clay content is close to 0 (e.g., < 1%), the clay content estimate is likely to be overestimated by a much larger fraction than would be the case for a larger clay content (e.g., 5% clay is 400% greater than 1% clay but only 50% lower than 10% clay). The result may be an upward bias in clay content for clay contents less than approximately 10%, which, according to Eq. (5), may be associated with channels less than 2-3 m in bankfull depth. Channels with bankfull depths less than 2 m (Sect. 2.1) were excluded from the analysis due to this potential bias."

I believe I have done the best I can with existing data to report and discuss all the primary sources of uncertainty, even if some of the sources of uncertainty are difficult to quantify (on bank angles especially) and thus preclude the propagation of a quantified uncertainty from the start to finish of the analysis. I have also tried to emphasize to the reviewer that, while the application of Eq. (5) (the old Eq. (4)) to specific locations is inherently difficult due to lack of local constraints on bank angle, the primary of objective of this paper is to document an increase, on average, in bankfull channel depth with increasing clay content. That objective has been met in a statistically comprehensive way.

I appreciate the reviewer's skepticism that broad-scale soil survey data can be used to parameterize clay content for point data. However, I note that soil survey data are derived from point-scale measurements (SCS scientists use data from soil pits to define soil properties across areas of similar slope, geology, etc.). Moreover, rates of bank retreat at a gaging station are likely controlled not by the clay content of a single point but rather by clay content over the spatial scale of a meander bend, which for large rivers can be ~1-10 km.

Q: "While I'm appreciative of the author's invoking the critical flow hypothesis (correct citation is Grant, 1997, not 2000 as it appears in text but not references), I'm somewhat confused by the role it plays in this story. As I understand it, the author argues that this hypothesis suggests that the range of Froude numbers for both sand and gravel-bed channels should be limited to near- or less than 1.0, and that therefore Froude number and discharge should be weakly and inversely related. The logic here is not entirely clear, and the mechanism that restricts Froude numbers is not entirely accurate. In its original form, the hypothesis argues that in steep channels (typically S>0.01), inter-actions between the free surface (particularly hydraulic jumps) and the bed result in a rough balance between forces that accelerate the flow and forces that extract energy from the flow and thus retard it, thus promoting near-critical flow conditions. This condition applies irrespective of grain size, although the author rightly points out that the actual bed features that set up this interaction are different for sand, gravel, and even boulder bed channels. A recent paper tests this idea in the flume and articulates the mechanism well (Piton and Recking, 2019). Most of the streams in the Dunne and Jerolmack data set have slopes much less than 0.01, and consequently much lower Fr, as the data shown in the paper point out."

A: I thank the reviewer for pointing out the incorrect citation, which has been corrected in the revised manuscript. The reviewer is correct that I generalized his 1997 critical flow hypothesis. The wording has been corrected in the revised paper. My reason for generalizing his hypothesis was that, even though the critical flow hypothesis was originally proposed for steep channels, the key idea (to my mind) of the critical flow hypothesis is the existence of a self-regulatory feedback in which an increase in velocity is met with an increase in drag that brings the velocity, and hence Fr, back down to a lower value. Similarly, a decrease in velocity tends to be met with a decrease in drag that tends to increase the velocity. It is likely that such a self-regulatory feedback is not limited to steep channels. In steep, gravel-bedded channels, faster flow tends to lead to more wave breaking and other drag-inducing processes, thereby tending to lower velocities back down to a critical value (i.e., Fr = 1). In less-steep, sand-bedded channels, faster flow tends to lead to the development of larger and more well-developed bedforms, which increases relative roughness and hence drag in the range of Fr values associated with well-developed bedforms ( $Fr \sim 0.1$ -1). The mechanisms for self-regulation differ between steep channels and less-steep channels, but selfregulation exists in both. In the revised manuscript, I clarify that the critical flow hypothesis applies to steep channels only and separately discuss evidence for a similar self-regulation of Fr in lesssteep channels.

**Text added on this point:**

"Grant (1997) proposed a critical-flow hypothesis in which depth-averaged water velocities are self-regulated via interactions between the water flow and the channel-bed morphology. Grant (1997) argued that, in steep ( $\geq 0.01 \text{ m m-1}$ ) channels, the Froude number rarely exceeds one for extended periods of time due to interactions between the free surface and the bed that result in an approximate balance between forces that accelerate the flow and forces that extract energy from the flow. Such a balance may also extend to coarse-bedded channels, the water flow in which is prone to wave drag associated with flow around bed sediment grains that protrude above the water surface (Wohl, 2013). Wave drag can be expected to be more common in coarse-bedded channels relative to channels with finer bed sediments both because they have relatively large bed roughness elements that more readily protrude above the water surface as well as a tendency towards shallower flows as a result of their characteristically large width-to-depth ratios (Schumm, 1960).

Central to the critical-flow hypothesis is the existence of a self-regulatory feedback in which an increase in velocity is met with an increase in drag that tends to reduce the velocity and hence the Froude number. Similarly, a decrease in velocity tends to be met with a decrease in drag that tends to increase the velocity. Here it is hypothesized that such a self-regulatory feedback is not limited to steep channels. In less-steep, sand-bedded channels, faster flow tends to facilitate the development of larger and more well-developed bedforms (which tend to form at Froude numbers ~0.1-1 (e.g., Simons and Richardson, 1966)) that increase relative roughness and hence drag. The mechanisms of self-regulation and the Froude numbers at which steep and less-steep channels may achieve this self-regulation thus differ, but both are likely to have self-regulatory interactions between the flow and the bed that limit the Froude numbers of bankfull discharges. Consistent with this hypothesis, here it is documented that bankfull Froude numbers, and hence

the ratio of depth-averaged water velocities to the square root of bankfull depths, tend to be within a relatively narrow range that has a weak inverse relationship to bankfull discharge."

Q: "More to the point, in my view, the primary control on Fr is channel slope, and I suspect that this is what is behind the weak inverse correlation between Fr and discharge (Fig. 3A). The simple theoretical dependency between Fr and S is shown in Grant (1997; ig. 4); a more sophisticated treatment is given in Palucis and Lamb (2017; Fig. 2). Since smaller streams tend to be steeper than larger ones, my hunch is that discharge is more of a proxy for slope than a driver as in Fig. 3A. This slope dependency is also probably lurking behind the grain-size/Fr relationship in Fig. 3B as well. None of this fundamentally invalidates the argument being made by the author but I think these relations should be acknowledged, as they have bearing on the physical mechanisms underlying the presentation."

A: Thank you for this comment. My work demonstrates how alluvial channel geometric variables interrelate. I used bankfull discharge as an independent variable because it has traditionally been used for this purpose. I did not and would not argue that any variable is more fundamental, significant, or a "driver" than any other variable. Bankfull Froude number depends explicitly on channel depth and velocity, the latter of which depends on slope and bed roughness. As such, bankfull Froude number certainly depends on channel depth, channel slope, and bed roughness (the latter of which depends on grain size and all of the factors that control bedform geometry (if bedforms are present)). I know of no basis for ranking the relative importance of any of these controls. This view is consistent with Palucis and Lamb (2017), who, despite relating specific channel morphologic types to specific slopes, document the importance of channel width and grain size in controlling channel type (e.g., "step-pools form in near supercritical flow or when channel width is narrow compared to bed grain size") and conclude that "certain bed slopes have unique channel morphologies because the process variables covary systematically with bed slope."

**Reviewer 2 (Roberto Fernandez):**

Q: "1. The use of the data by Dafalla (2013) seems misrepresented in this manuscript (Figure S1). Cohesion values reported by Dafalla (2013) correspond to pure sand, sand and clay mixtures with 5

In addition, Dafalla (2013) shows that for the same clay content (15

I really believe that the approach proposed in this manuscript has a lot of potential and others have included geotechnical considerations in models for stream restoration (e.g. CONCEPTS, see Langendoen et al. 2001; RVR Meander, see Motta et al. 2012). I would encourage the author to dig out some more references regarding cohesion estimates for soils that are more relevant for the Mississippi River Basin. For example, Masada (2009) presents a very extensive report on geotechnical parameters for the state of Ohio and includes different relations between sediment properties (cohesion for example) and soil composition (amount of silt, amount of clay, etc.). Their approach is specific to highway embankments but the results of their tests might be more general in terms of cohesion values in relation to clay contents."

A: I greatly appreciate the reviewer's drawing my attention to these additional sources of data, Masada (2009) in particular. I have scoured the literature for additional sources of data but have not found any. Masada (2009) is not a good source of data on the relationship between cohesion and clay content because he presents cohesion versus clay content for a very narrow range of clay contents (e.g., 27%-34% in Fig. E6). Also, I prefer to use data from direct shear tests rather than the triaxial shear tests of Masada (2009) because direct shear tests are more similar to gravitational shear failures. Dafalla (2013) is also a good source of data because he presents data for cohesion versus clay content at fixed moisture contents. Assuming that there is no systematic variation between short and tall banks in the water content at which channel banks tend to fail, it is the dependence of cohesion on clay content at similar moisture contents (or averaged across all moisture contents) that is most appropriate for use in the model.

The reviewer is correct that the sand and clay mixtures considered by Dafalla (2013) were obtained by mixing sand and clay as opposed to being natural soils. However, I don't think creating such mixtures necessarily undermines the quality of the data or the applicability of the results to natural soils. Masada (2009) considered natural soils but they were almost entirely glacial in origin, hence not necessarily a better analog for alluvial soils than the mixtures studied by Dafalla (2013).

Text added on the Dafalla (2013) data:

"Bank material cohesion varies linearly from 0 (for cohesionless sand) to approximately 90 kPa (for pure clay) for moisture contents in the range of 7 to 40% according to a least-squares linear regression of the data from Dafalla (2013) (Fig. S1):

$$C \approx (900 \pm 70) p_{\rm c}.$$
 (3)

where the units of C is Pa and pc is percent. The uncertainty in Eq. (3) is the standard error (i.e., one standard deviation) resulting from the regression."

and the caption for Figure S1 provides some additional technical information:

"Figure S1. Plot of cohesion, C, as a function of percent clay content, pc, from the experiments of Dafalla (2013) demonstrating the approximately linear nature of the relationship, i.e., the exponent of a power-law relationship between C and pc (shown as a line above) determined by a least-squares linear regression to the logarithms of the data is  $0.78 \pm 0.12$  where 0.12 is the standard error. Note that three data points from Dafalla (2013) were not included because the clay contents were zero and hence the logarithms were undefined. However, those data points were included in the linear least-squares regression of non-log-transformed data reported in the paper that resulted in the coefficient of proportionality of 900  $\pm$  70 in Eq. (3)."

Q: "2. The use of equation (2) might not be appropriate for riverbanks. The use of that equation as presented by Chen (1969) and Terzaghi et al. (1996; p. 271-272) is for soil embankments located above the water table. Several authors have used it in the past as discussed by ASCE (1988) but even there, the authors suggest that critical depth approaches are not accurate when the most common bank failure mechanisms for riverbanks are due to tension cracks that cause toppling or cantilever failures. Assuming the equation is indeed an appropriate approach for riverbanks, I would en-courage the author to explore the sensitivity of its input variables to other values. Chen (1969) shows a wide range of Ns values that depend on the internal friction angle of the material (which is sensitive to moisture content) and the actual slope of the bank. The smallest stable bank height would be given by the smallest possible safety param-eter Ns so why not explore a range of Ns values. When the channel has low flow, the bank might be quite dry and its maximum stable height would be quite different from that obtained with a saturated bank (e.g. during the falling limb of a hydrograph where the river stage is getting lower but the bank remains saturated). It would be very useful to see these considerations in the analysis. The author discusses the issue briefly but more details regarding bank failure mechanisms and their prevalence might strengthen the manuscript."

A: The reviewer is correct that the analysis of Chen et al. (1969) is for unsaturated banks only. I have augmented the results of Chen et al. (1969) with other studies to provide a more accurate basis for the 0.35 proportionality coefficient between percent clay content and critical bank height. Text added on this point:

"The maximum stable height, hc, of an alluvial channel bank subject to gravitational shear failure is proportional to bank-material cohesion, C (Taylor, 1937; Terzaghi and Peck, 1967; Hunter and Schuster, 1968; Chen et al., 1969; ASCE, 1999):

$$h_{\rm c} = \frac{N_s}{\rho q} C, \tag{2}$$

where  $\rho$  is the bulk density of the bank material, g is the acceleration due to gravity, and Ns is a stability parameter dependent on the geometry of the potential failure surface (e.g., planar, log-spiral, or circular), the pore pressure of the bank material (which is governed by the water table position if the pore pressure is assumed to be hydrostatic), and the angles of the bank and of internal friction (see Table 1 for a list of variables).

In order to estimate a reference Ns value appropriate for understanding how gravitational stability may influence the scaling of alluvial channel bankfull depths to discharge, a steep bank (i.e., near-vertical at the top of the bank but decreasing to approximately 45° near the toe) with an internal friction angle of 35° (typical for a loamy or clayey sand), near-saturated conditions, and a log-spiral potential failure surface were assumed. Near-saturated conditions are consistent with the fact that gravitational shear failure has been documented to occur most frequently during the falling limbs of flood discharges when pore pressures tend to be associated with near-saturated conditions (e.g., Casagli et al., 1999; Simon et al., 2000). Chen et al. (1969) derived Ns values for prescribed angles of the bank and of internal friction for unsaturated conditions. For a friction angle of 35°, Ns values in Table 1 of Chen et al. (1969) decrease with increasing bank angle from Ns = 22 for a  $60^{\circ}$  bank to Ns = 12 for a 75° bank and Ns = 7.5 for a vertical bank. Hunter and Schuster (1968) limited their analysis to cases with no internal friction (hence their absolute Ns values are not applicable here) but documented an approximately 3-fold decrease in Ns values from unsaturated conditions (i.e., M = hwyw/hcy'  $\approx$  1, where is hw is the depth to the water table below the top of the bank,  $\gamma$  w is the unit weight of water, and  $\gamma'$  is the submerged unit weight of the bank material) to near-saturated conditions (i.e., M = 0). Combining the results of Chen et al. (1969) and Hunter and Schuster (1968) suggests that Ns values for a saturated bank with an internal angle of friction of 35° vary from approximately 7.3 for a 60° bank to Ns  $\approx$  4 for a 75° bank and 2.5 for a vertical bank.

Bank material cohesion varies linearly from 0 (for cohesionless sand) to approximately 90 kPa (for pure clay) for moisture contents in the range of 7 to 40% according to a least-squares linear regression of the data from Dafalla (2013) (Fig. S1):

$$C \approx (900 \pm 70) p_{\rm c}.$$
 (3)

where the units of C is Pa and pc is percent. The uncertainty in Eq. (3) is the standard error (i.e., one standard deviation) resulting from the regression.

Combining Eqns. (2) and (3) and assuming a bulk density of 1700 kg m-3 and a representative value of Ns  $\approx$  6 (corresponding to a near-saturated bank with an angle of approximately 65°, i.e., an average angle for a bank that is near-vertical at the top and decreases to an angle of approximately 45° near the toe) yields

$$h_{\rm c} \approx 0.35 p_{\rm c}.$$
 (4)

Absent site-specific data for bank angles, the largest source of uncertainty in the proportionality coefficient in Eq. (4) as applied to specific locations is likely the bank angle, since relatively modest variations in bank angle (e.g., from  $90^{\circ}$  to  $75^{\circ}$ ) are associated differences in Ns of approximately a factor of 3 while other sources of uncertainty (e.g., between cohesion and clay content as quantified by Eq. (3)) are smaller. Section 4 provides discussion on how uncertainty in bank angle and other factors such as bank vegetation limit the precision of Eq. (4) to specific locations. The primary of objective of this paper, however, is to document an increase, on average, in bankfull channel depth with increasing clay content:

 $h \approx 0.35 \, p_c. \tag{5}$

assuming that bankfull depth is approximately equal to the maximum gravitationally stable bank height."

Q: "3. Sensitivity analysis: Figures 2b and 2c present results for bank heights based on a synthetic dataset. If the author estimated clay contents using averaging windows for a soils dataset, why not extract second order statistics from it and use them directly instead of creating a synthetic dataset?"

A: I don't think that second-order statistics (e.g., coefficient of variation) necessarily capture all of the potential uncertainty in a dataset. For example, spatial variations in the values of a dataset that systematically over- or under-predict actual values won't correctly capture the true uncertainty. I also think creating a synthetic dataset has advantages. For example, a key goal was to demonstrate that an upward bias can exist in very low clay contents. I don't think using second-order statistics in the averaging would necessarily have the same potential to demonstrate this phenomenon.

Q: "4. Use of the Mississippi River Basin data: The author clearly states why the MRB data are used. However, not knowing much about the many different locations along the basin, I have a few questions. (1) What percentage of the cross sections analyzed can be considered natural? (2) Did the author discard those locations where the navigable channels are maintained by the US Army Corps of Engineers? (3) Of the many stations used, how many might be influenced by river control structures (dams, wing dams, chevrons, etc.) or road infrastructure (e.g. culverts, bridges)?"

A: Using Google Earth, I have examined the locations of the 387 stations and have found no instances where cross-sections are located close to infrastructure. This lack of overlap between the U.S.G.S. station locations and infrastructure may be partly due to the fact that my filtering criteria (lines 88-92) did an effective job at removing stations with stage-discharge relationships that are affected by infrastructure.

Q: "5. Figures 3 and 4: It is not at all clear why the author includes regression plots of the Dunne and Jerolmack (2018) dataset. Based on the abstract and introduction, it was unexpected that a different dataset appears in the manuscript and becomes the focus of the second half. I understand the use of the dataset for Figure 5, which is new but the content of Figures 3 and 4, is not. I would encourage the author to make it clear to the reader earlier that the DJ dataset is a substantial part of the analysis and to state explicitly the novelty of including figures 3 and 4." A: The opening sentence of the abstract makes clear that the problem I am tackling is the powerlaw scaling of bankfull depths, widths, depth-averaged water velocities, and along-channel slopes to bankfull discharge in alluvial channels. The first part of my paper deals with control on channel depth only, so it stands to reason that there must be another part of the paper that extends the work on channel depth to other aspects of channel geometry using additional principles. Figures 3 and 4 (including the reporting of best-fit exponents) have not been published elsewhere and their inclusion is important for meeting the goals of the paper.

Q: "6. Figure 5: I have a few specific questions about the analysis leading to Fig. 5. (1) What is the number (and percentage) of cases that report ripples/dunes over the entire Ohata (2017) dataset? (2) For those reporting ripples/dunes, what is the number and percentage of measurements obtained in the laboratory and in the field? (3) For those in the field, how many are for large rivers? Cisneros et al. (2020) show that traditional dune scaling equations overestimate the size of dunes in large rivers and propose the following relation between dune height (H) and water depth (h) – H 0.056h - 0.12h. 4) Are the only sources of roughness in the DJ data the ripples/dunes or gravel size? What about bars, meandering, vegetation?"

A: 1) 1574 (42%) of the 3790 data points in Ohata et al. (2019) have ripples or dunes (noted in revised manuscript), 2) of that 42%, 19% are in the field and the remaining 81% are in the laboratory (noted in revised manuscript), 3) 3.2% (123 out of 3791) of the data points are from rivers with h > 5 m. 4) I am assuming that the dominant (not only) sources of roughness in the channels of the DJ data are ripples/dunes or gravel clasts. Long-wavelength topographic features such as bars and meanders are not likely to be dominant roughness/drag-inducing elements given that the presence/absence of the flow separation that tends to dominate drag depends sensitively on the maximum slope of bedforms and other obstacles to the flow, with slopes in excess of 0.2 m/m generally needed to trigger the flow separation (though surface curvature also plays an important role in addition to slope; see below). I concede that vegetation can be a dominant source of roughness on the beds of ephemeral channels, and some of the scatter in my analysis may be a result of vegetation-induced bed roughness.

Text added on this point:

"Long-wavelength topographic features such as bars and meanders are not likely to be dominant roughness/drag-inducing elements given that the presence/absence of the flow separation that tends to dominate drag depends sensitively on the maximum slope of bedforms and other obstacles to the flow, with slopes in excess of 0.2 m/m generally needed to trigger flow separation (e.g., Lefebvre et al., 2014). Vegetation can certainly be a dominant source of roughness on the beds of ephemeral channels, however, and some of the scatter in the analysis of this paper may be a result of vegetation-induced bed roughness."

Cisneros et al. (2020) demonstrates that the lee-side angle of many bedforms in large alluvial channels is lower that empirical equations predict and argues that such lower angles means that flow separation and hence drag is less significant than empirical models would suggest in large rivers. It is important to note, however, that flow separation depends sensitively on the surface curvature in the zone of the maximum adverse pressure gradient (Lamballais et al., 2010), not just on the relative height or lee-side angle of bedforms. As such, more research is needed to conclude that flow separation is rare on the lee sides of bedforms in large alluvial channels.

Q: "As a final general comment, I was hoping to see more analysis on the Mississippi River Basin dataset and comparisons between it and the DJ dataset where possible. The manuscript seems to be split between two separate analyses but the abstract and introduction do not suggest that. I recommend the author to modify these initial sec-tions as necessary and compare the MRB data with the DJ data where possible. What kind of relation does the author obtain between bankfull depth and bankfull discharge for the MRB under the geotechnical considerations? On the other hand, could clay contents (and cohesion) be estimated with a revised version of equation (4) for other rivers in the world where soil data is not readily available?"

A: See my response to point 5 on the apparent split in the manuscript. It should be possible to infer clay content values for some rivers *if bank angles and moisture contents were also well constrained*. We do not have such data yet for many alluvial channel cross sections across a sizable region.

Q: "1) How do the bankfull estimates found here for the MRB compare to those of Dong et al (2019). This reference appears in the introduction but is not mentioned in the discussion. A: Since Dong et al. (2019) deal with the Selenga River Delta and my paper deals with the MRB, it is difficult to make a direct comparison with their results.

Q: "2. I did not understand the fourth criteria used to keep a USGS gaging station in the analysis of the MRB."

A: I apologize that this was not clearer. The estimation of bankfull stage requires fitting stage to discharge for the smallest and largest values. The bankfull stage is estimated to be where these two lines meet. In order to verify that the low-flow fit is reasonable, I retained only those stations for which the extrapolation of the fit of low-flow values passes "close" to the correct value: zero flow depth at zero discharge. In cases where the extrapolation of the fit does not pass close to zero, the data are likely of insufficient quality or require an adjustment based on data that are not publicly available. What represents "close" should not be based on an absolute error, e.g., 0.5 m, because such a criterion would require that low-flow fits for a deep channel be much more accurate than one for a shallow channel. So, instead, I defined "close" as within 50% of the bankfull stage from zero. That is, if the bankfull stage is 5 m, then the extrapolation of the low-flow fit to zero discharge must yield a stage within 2.5 m of zero. Similarly, if the bankfull stage is 2 m, then the extrapolation of the low-flow fit to zero discharge must be within 1 m of zero.

Text added on this point:

"The analysis presented here began by including data from all U.S.G.S. gaging stations in the MRB with available peak discharge data (U.S. Geological Survey, 2020). Only those stations for which the slope of the high-flow linear regression is at least five times smaller than the slope of the low-flow linear regression were retained. In addition, only those stations that had at least 20 years of data, have a contributing area larger than 100 km2, are not located close to major infrastructure (based on an inspection of each station location in Google Earth imagery), and have a resulting bankfull depth of greater than 2 m were retained. Channels with bankfull depths less 2 m were removed because such channels tend to be associated with low clay contents that are inherently difficult to estimate in the field (see Sect. 3.1 for more detail on the potential bias associated with estimating low clay contents). In order to further filter out stations where the low-flow linear regression is potentially unrepresentative of the hydraulic behavior of in-channel discharges, only those stations for which the extrapolation of the low-flow linear regression is close to zero flow

depth at zero discharge were retained. Stations for which the low-flow regression does not extrapolate to a flow stage close to zero at zero discharge may have gage height data that are not an accurate proxy for flow stage and/or have other data quality issues that preclude an accurate estimate of bankfull depth using the stage-discharge rating curve. What represents "close" should not be based on an absolute error, e.g., 0.5 m, because such a criterion would require that the low-flow regressions for deep channels be relatively more accurate than those for shallow channels. Here "close" was defined as being within 50% of the bankfull stage from zero. That is, if the bankfull stage is 5 m, then the extrapolation of the low-flow regression to zero discharge must yield a stage within 2.5 m of zero in order for that station to be retained in the analysis. Similarly, if the bankfull stage is 2 m, then the extrapolation of the low-flow fit to zero discharge must be within 1 m of zero. A total of 387 stations met these criteria."

Q: "3. If the analysis discards rivers with depths smaller than 2m, why is the 0.5m to 1.5m soil depth the only section considered for the analysis. What about river sections with different bank layers? The author mentions that soil data below 1.5m is not reliable but how valid is it to assume a uniform soil profile for the entire channel depth? How sensitive is the proposed model to this assumption?"

A: I assume that the texture of the top 1.5 m of the bank (for which data is readily available) is representative of the entire bank. I understand that this assumption may be violated in many cases. However, given that the floodplain deposits that comprise many banks are the depositional products of channels where well-sorted sediments tend to be the norm, it is reasonable to expect that the texture of the uppermost 1.5 m of the bank will correlate strongly with bank material at greater depths in many cases.

Q: "4. *Line 150(and other locations) - Ohata et al 2017 (not 2019).*" A: That you for pointing out this typo. It has been fixed in the revision.

*Q*: "5. Line 151 "How did the author "cross-reference" the Ohata et al. dataset with the Dunne and Jerolmack dataset?

A: Cross-referencing refers to the development of a curve in Fr vs.  $d_{50}$  space that separates channels that have ripples and/or dunes from those that do not, and using that curve to infer that the vast majority of sand-bedded channels in the Dunne and Jerolmack dataset have ripples and dunes. This is done by assuming the existence of ripples and dunes in channels of the D&J dataset that have Fr and  $d_{50}$  values that sit above and to the left (Fig. 3B) of the envelope curve separating channels with and without ripples and dunes in the Ohata et al. (2017) dataset. Text added on this point:

"By cross-referencing those results with the Dunne and Jerolmack (D&J) (2018) global dataset (i.e. by drawing a curve in F vs.  $d_{50}$  space that separates channels that have ripples and/or dunes from those that do not, and assuming the existence of ripples and dunes in channels of the D&J dataset that have F and  $d_{50}$  values that sit above and to the left (Fig. 3B) of the envelope curve separating channels with and without ripples and dunes in the Ohata et al. (2017) dataset), Sect. 3.3 demonstrates that 96% of sand-bedded channels in the D&J global dataset have F and  $d_{50}$ values conducive to ripple and/or dune development and therefore have a roughness that is likely dominated by bedforms rather than by bed sediment grains." *Q*: "6. Line 214 "what is the equation of the curve (envelope) used to identify the conditions conducive to dune/ripple development?"

A: There is no equation. This is simply a drawn curve. Noted in revised manuscript.

Q: "7. Line 259 (and other locations) "Chen 1969 (not 1971)"

A: Typo fixed.

*Q*: 8. Line 286 "Vegetation. What are the predominant types of vegetation along the MRB? How deep are their roots? Root length might set slump block thickness. Vegetation might be the most relevant factor in shallow channels (up to max root length) and geotechnical considerations might be more relevant in deeper channels where roots might not stabilize the full bank.

A: The stability of any bank is primarily a function of its weakest portion or layer, i.e., failure is more likely to occur in a zone of low shear strength compared to a zone of high shear strength, all else being equal. Failure of a weaker zone underlying a stronger zone can create a cantilever, but such cantilevers cannot continue to grow indefinitely, hence rates of long-term retreat in stronger and weaker zones of the same bank will tend to be similar. This, together with the fact that vegetation is likely to strengthen just the uppermost approximately 2 m of channel banks (globally, >99% of roots are found in the top 2 m of the soil (Jackson et al., 1996)), suggests that bank material shear strength is not likely to be controlled primarily by vegetation in channels deeper than 2 m. I concede that some plant roots can exceed (even greatly exceed) 2 m in depth. However, plant roots become quite small in density at > 2 m depth based on Jackson et al. (1996). I understand that this view appears to contradict the dependence of bank retreat rates on vegetation that has been documented in many studies (e.g., Ielpi and Lapôtre, 2020). I am not stating that vegetation has no effect on bank stability. Rather, I am stating that I know of no study that has attributed bank shear strength to vegetation that has accounted for the fact that wetter climates with more vegetation also tend to have more clay-rich soils. As such, much of the apparent control of vegetation could be due to clay content.

Text added on this point:

"Bank vegetation plays a significant role in controlling bank stability, but it is unlikely that such control is responsible for the scaling relationships that are the focus of this paper, as such scaling relationships exist across climatic regions with very different vegetation characteristics. In addition, the stability of any bank is primarily a function of its weakest portion or layer, i.e., failure is more likely to occur in a zone of low shear strength compared to a zone of high shear strength, all else being equal. Failure of a weaker zone underlying a stronger zone can create a cantilever, but such cantilevers cannot continue to grow indefinitely, hence rates of long-term retreat in stronger and weaker zones of the same bank will tend to be similar. This, together with the fact that vegetation is likely to strengthen just the uppermost approximately 2 m of channel banks (globally, >99% of roots are found in the uppermost 2 m of the soil (Jackson et al., 1996)), suggests that bank material shear strength is unlikely to be controlled primarily by vegetation in channels deeper than 2 m. A dependence of bank retreat rates on vegetation has been documented in many studies (e.g., Ielpi and Lapôtre, 2020). However, such studies may not fully account for the fact that wetter climates with more vegetation also tend to have more clay-rich soils, leaving open the

question of whether it is bank vegetation or material texture that is most responsible for bank resistance to erosion."

**Reviewer 3 (Christopher Hackney):**

Q: "I find the section between lines 30 and 34 quite confusing. The author states that channel incision or floodplain deposition may increase bank height (OK so far). This causes banks to collapse once a critical bank height is reached. The subsequent failure results in channel widening which tends to increase water depths back towards the stable bank height. It is this last bit that I find counter to the previous few lines. I follow the argument that a wider channel results in slower flows, but the reduction in flows is a result of the increased channel capacity due to increasing width, and so there is no need for the channel to continue to incise of raise water depths to hold the same volume of water, as the increased width accounts for this. Furthermore, it has been shown recently that over long time frames, channels tend to maintain an equilibrium width (and thus presumably bank height; see Mason and Mohrig, 2019) and that the channel adjustment due to bank collapse also often sees increased deposition on the inner bank (see van de Lageweg et al. 2014). A clearer explaination of how bank failures can result in increased water depths would be welcome."

A: The reviewer is correct that the sentence "Bank failure results in channel widening, which may reduce depth-averaged water velocities and therefore tend to increase water depths (to convey similar water discharges) back towards the maximum stable bank height" was confusing and did not accurately convey my conceptual model for how alluvial channels self-regulate to a maximum depth comparable to the maximum stable bank height. I have replaced it with: "The gravitational failure of channel banks may partially control bankfull depths via a self-regulatory mechanism in which channel incision and/or floodplain deposition tend to increase bank height, triggering bank failure when a critical bank height, dependent on bank material cohesion, is exceeded (Andrews, 1982), thus introducing new sediment into the channel bed that, as it is redistributed by fluvial processes, tends to reduce the channel depth back towards a critical value."

Q: "The author also notes a few potential limitations to the data exploration in the discus-sion, but does so briefly and in passing. I would like to see a more developed discus-sion around the role of vegetation induced cohesion, and also the role of failed material (particularly in clay rich soils which are more likely to fail and persist in blocks at the base of the river bank), as these are likely to be key local controls on any variation in the relationships the author has presented. Another potential source of variaiton that isn't raised but may also be important is the role of floodplain topography in defining bankful depths and bank heights. If a bank is eroding through a scroll-bar then it is likely that following a failure the local bank height may decrease as a result of variable local topography (i.e. on a floodplain where elevation is sloping away from the bank). Following the model presented here, will that new bank remain stable until its critical height either is reached through channel incision or build up from floodplain deposition? Following on from this, on Ln 272 the author states that an increase in bank height caused by floodplain deposition may trigger bank failure. Presumably, to deposit ma-terial on the bank the flow needs to be over-bank. Therefore, is it the deposition of the material on the bank during these flows, for the increased water velocity and bank shear stresses that will induce this erosion?"

A: Discussion items are, by nature, relatively brief (in that they are limited to subsections of the Discussion section only). To the extent that my discussion items are relatively brief, I think this is appropriate given that there is no indication that vegetation-induced cohesion causes the scaling of alluvial channel geometry that is the subject of this paper. Also please see my response to reviewer 2's comment 8 on the issue of vegetation.

The revised manuscript mentions the role of failed bank material in the context of the interplay between fluvial scour and gravitational failure in contributing to channel width adjustment.

Regarding the final question, either incision or floodplain deposition can trigger bank failure: it is not one or the other. Whether a channel incises depends on changes in base level and sediment supply that depend on tectonic and climatic processes that operate at scales much larger than a channel reach. A channel that is relatively shallow relative to its bankfull depth will tend to flood overbank more often than a deeper channel, thereby promoting deposition until the channel deepens to the point where channel banks fail. The model of this paper invokes a long-term dynamic equilibrium between channel deepening processes (channel incision and/or floodplain deposition) and channel shallowing processes (bank retreat, which introduces sediment into that channel that, once redistributed laterally, causes channel aggradation). My response to the reviewer's first concern clarifies (I hope) this aspect of the conceptual model in the revised manuscript.

Q: "Overall, I think a lot of the issues raised above come down to the temporal scale being examoined here and there is a need for some discussion the manuscript around the ime-scales over which these processes may become the dominant factor and how time-averaging of the other processes involved in river bank erosion occurs. Could the author examine any historical rates of bank erosion for sites analyses in the manuscript to see whether the theoretical model holds for different time periods?"

A: I appreciate this suggestion. However, while rates of bank erosion are available over multiple time scales, what matters to this paper is channel widening. Channel widening likely occurs most abruptly during channel incision periods (following climatic changes, for example), after which bank retreat continues but channel widening does not because cut bank migration is approximately balanced by point-bar progradation (e.g., Mason and Mohrig, 2019). I am not aware of any available data on channel widening data over multiple time scales. I also wish to emphasize, as I did in my responses to reviewer 1, that I am not arguing that gravitational failure is the dominant processes setting channel depth and width. Fluvial scour and gravitational failure influence and complement one another so intimately that it is difficult to clearly identify a dominant control.

Thank you,

JmD. Pelt

Jon D. Pelletier Professor jdpellet@email.arizona.edu

[revised manuscript text omitted]

---

## Referee Report (RR1)

**Review of: Controls on the hydraulic geometry of alluvial channels: bank stability to gravitational failure, the critical-flow hypothesis, and conservation of mass and energy by Jon D. Pelletier**

**Roberto Fernández**

**Jan. 29th, 2021**

I have read the author's response to all reviewers and the new manuscript and believe the paper is ready to be published. The author has offered answers to issues raised and edited the manuscript accordingly to improve its clarity. I would have liked to see more references with work such as that by Dafalla (2013) to strengthen the basis for the relation between clay content and cohesion beyond the small subset of clay/sand materials used therein. I see this as the weakest element of the analysis but finding more of such data might be something the author could also include as suggestions for future work. I have included some minor things below and hope to see this paper published soon.

Line 43 – unclear sentence. Should it read:

"Such a balance may also extend to coarse-bedded channels, in which the water flow  is prone to wave drag…"?

Line 58 – requirement or consequence?

The bankfull widths of alluvial channels are set by  geomorphically effective water discharges.

Line 102 – Extra 'of' in sentence

The primary  objective of this…

Discussion needs some revision.

Line 339-342 mentions two model simplifications: tension cracks and vegetation.

Line 355 onwards mentions four additional assumptions with the first one being tension cracks mentioned as one of the two model simplifications above.

Merge and call them 5 assumptions/simplifications?

Line 392-393 – Li et al. (2015) not 2005.

Line 406-408 – A quick mention of the research needs identified in Section 4 would make the conclusions stronger (and also easier for people who might not have time to read the full paper).

Line 412 – Roberto Fernández not Hernandez (Alt+160 for **á**). Thanks!

---

## Author Response (AR2)

I wish to the reviewers for their careful review of the revised manuscript. I especially wish to thank Roberto Fernández for his careful reading. I have adopted all of this suggestions except for one. I chose to retain the sentence "The bankfull widths of alluvial channels are set by the requirement that channels convey geomorphically effective water discharges" because I think this more accurately conveys the idea I want to express than his suggested alternative.

[revised manuscript text omitted]